# Earthen Jewish Architecture of Southern Morocco: Documentation of Unfired Brick Synagogues and Mellahs in the Drâa-Tafilalet Region

**Eva Matoušková [1], Karel Pavelka [1,\*], Tobiáš Smolík [2] and Karel Pavelka Jr. [1]**

[1]  Faculty of Civil Engineering, Czech Technical University in Prague, Thakurova 7, 16629 Prague, Czech Republic; eva.matouskova@fsv.cvut.cz (E.M.); karel.pavelka@hotmail.com (K.P.J.)

[2]  Faculty of Arts, Charles University in Prague, Jan Palach Square 1-2, 11638 Prague, Czech Republic; tobias.smolik@post.cz

\*  Correspondence: pavelka@fsv.cvut.cz; Tel.: +420-608211360

**Abstract:** This article seeks to highlight the vanished and not-so-well-known material culture of historical southern Moroccan Jewry. Jewish settlements could be found practically in the whole of North Africa before the Second World War; however, afterwards, it almost completely disappeared due to the political changes in the region and the establishment of the state of Israel. In southern Morocco, the last Jewish communities were present until the 1950s. Thanks to the interest of the Moroccan authorities, an effort has been made to restore some monuments and keep them as part of the cultural heritage that has attracted foreign tourists for the last few years. As part of the expeditionary research of Charles University and the Czech Technical University in Prague, several documentation projects were carried out in 2020, some of the results of which are described in this paper. Modern automatic methods of geomatics, such as easy to use laser scanning, mobile laser scanning in PLS modification (personal laser scanning), and close-range photogrammetry were used. The results of documentation were processed in the form of 3D models and basic plans, which are used mainly for analyzing residential zones of the Jewish population, the so-called *mellahs*. In this article, two case projects are described. In both cases, all the mentioned documentation methods were used. The technologies used were analyzed in terms of data collection speed, price, transport, and possible difficulties in use. The PLS technology is relatively new and still under development, such as miniaturising of other measuring instruments. Accuracy testing and usability of above-mentioned technology in cultural heritage documentation real practice is the benefit of this research. Finally, a second aim was to provide information of abandoned cultural places and constructions, which are on the edge of interest and endangered by destruction. It clearly shows that PLS technology is very fast and suitable for these types of objects.

**Keywords:** Morocco; Moroccan Jews; *mellahs*; cultural heritage; Jewish architecture; 3D documentation

## 1. Introduction

### 1.1. The Jews in Morocco

The Jews of southern Morocco define their identity in contrast to the Jews expelled from formerly Muslim Spain. Contemporary researchers assumed the existence of functional Jewish communities even 200 years earlier, before the arrival of Islam [1]. The valleys of the rivers Drâa, Sous, and Ziz, connected to the Trans-Saharan trade routes [2], are considered to be some of the oldest areas of Jewish settlement [3]. In the South Moroccan oases, the Jews worked in small settlements of a more rural character, where they were in close contact with the majority Berber population. Even though today Moroccan Jews live in small numbers only in some towns north of the Atlas Mountains, in the southern area there is still a hidden wealth of Jewish monuments. Synagogues,

public buildings, and entire Jewish quarters that are either inhabited and secondarily used or abandoned and due to weather conditions subjected to irreversible destruction. These are the last extensive documents of Jewish architecture made of unfired brick that document the archaic continuous Jewish settlement in the Islamic world.

### 1.2. Jewish Architecture in Morocco

The synagogues and Jewish quarters (*mellahs*) of southern Morocco are a unique cultural and architectural phenomenon (Figures 1 and 2). These distinctive products of folk architecture speak of the coexistence of different cultures, their dialogue, and intermingling. The remains of the original community of Moroccan Jews, who made up 2.5% of the Moroccan population in the first half of the 20th century [4], have left the southern regions completely. The term *mellah* stands for a demarcated and often walled Jewish quarter with its own gates. Study of the Jews in a region of Atlas Mountains and Sahara is very limited from the lack of resources. Due to their cultural and historical difference from the North Moroccan Jewry, they are often dealt with separately and related to other Jews of the Sahara region, while a comprehensive monography on this topic is still missing. While the Amazigh people (Berbers), as majority in the desert areas, do not possess almost any written historical primary sources, the history of the southern Moroccan Jewry is therefore derived from the (sometimes very tendentious) travelogues of foreign explorers. The problematic character of these sources also facilitates the persisting of some myths about "Berber Jews". Therefore, we avoid this term and we do refer to our subject solely as the "Jews of Southern Morocco" [5,6].

Abandoned structures made of unfired clay thus decay irretrievably and disappear with the rain. Nevertheless, this interesting phenomenon has not yet received adequate attention or systematic research and documentation [7]. The only systematic research on Moroccan synagogues containing few earthen architectures was conducted by Joel Zack in the 1980s. However, most of the documented sites were urban synagogues in northern Morocco, and except for an abridged report, the study was not published. There are also a few earthen synagogues photographed as part of the project "Diarna: The Geo-Museum of North African and Middle Eastern Jewish Life" [8].

The area of southern Morocco is made up of rural regions that have long faced poverty, urbanization, and migration to the economically stronger part of the country [9]. Slowly, the developing infrastructure and emigration, unbalancing natural growth, conditioned the state of urban units.

Nevertheless, the local population is faithful to traditional materials and continues to inhabit and maintain archaic clay estates. In many cases, the former Jewish quarters are inhabited or secondarily used for storage and agricultural purposes. Newly and spontaneously constructed individual buildings are made of unfired bricks, concrete, or combined materials. If a systematic construction is created from modern materials, it usually happens outside the historic core of the village, which is not yet systematically demolished. This is made possible by both the character of the landscape and the character of clay castrates. The density of buildings and the interconnection of building units and roofed alleys do not allow for a gradual transformation into an urban system.

The main factor that is dynamically changing the shape of Morocco, including its Jewish cultural heritage, is general tourism. The imposing urban synagogues and the former Jewish quarters in the northern part of the country are popular tourist destinations.

In the southern rural areas of the country, mass tourism, with few exceptions, still has minimal impact. However, the Moroccan tourism industry is one of the fastest growing on a global scale [10], so the expansion of tourist zones towards the Sahara cannot be ruled out in the future. The areas south of the Atlas are already an integral part of excursion itineraries, but due to the lack of infrastructure and the limited number of established tourist destinations, it is largely limited to a shuttle service to the popular desert dunes and back. There is a possibility for the future, that the earthen architecture,

Jewish, Berbers, and Muslims, will interesting for tourists. Therefore, it is necessary to document and save it.

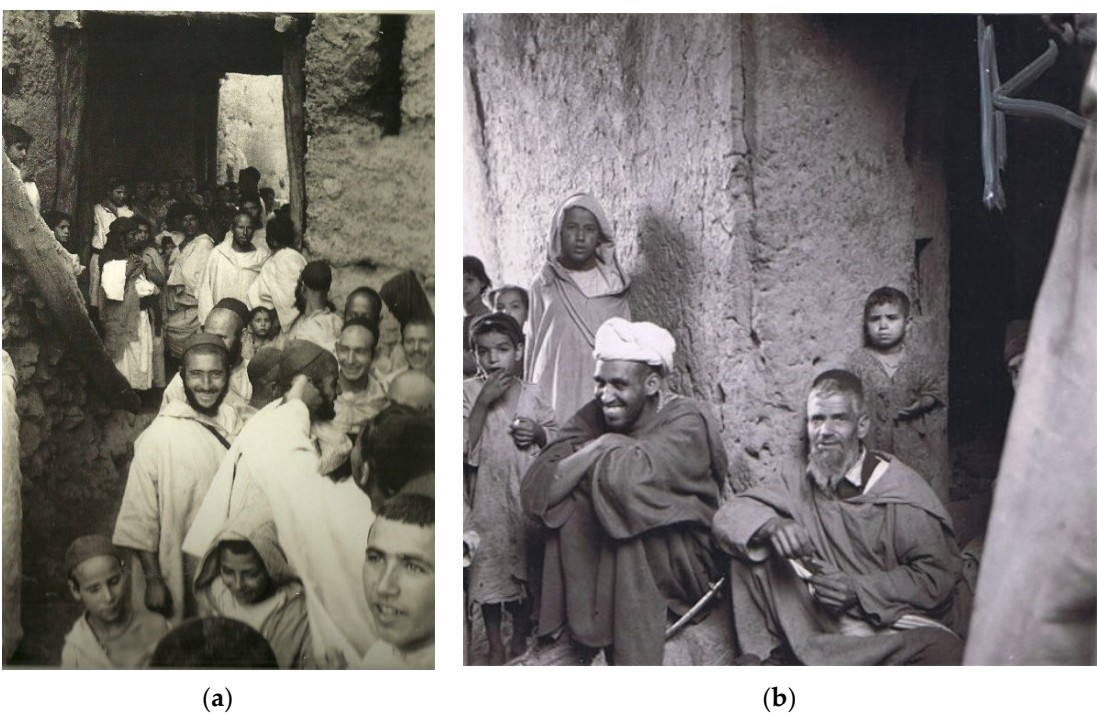

(**a**)　　　　　　　　　　　　　　　　　　(**b**)

**Figure 1.** (**a**) Ait Bou Oulli *mellah* (photographer unknown), (**b**) Amezrou *mellah*, 1950 (Schulmann, Zédé), https://www.moroccan-judaism.org/phototheque. Paul Dahan collection.

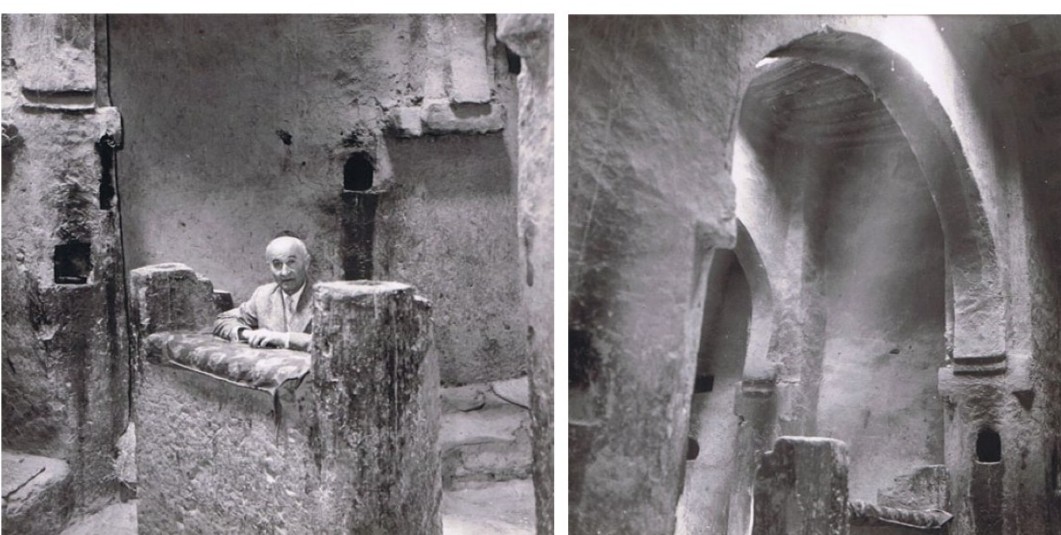

**Figure 2.** Synagogue of Amezrou, 1950 (Schulmann, Zédé), https://www.moroccan-judaism.org/phototheque. Paul Dahan collection.

The local population maintains the narrative of a smooth coexistence, with the image of Jews as close neighbors and collaborators, different in certain customs, but with equivalent expressions. An emphasis is placed on the view of Jews as intermediaries between individual homesteads and villages and the countryside, essential links in the chain of economic relations based on trading. The cultural heritage of the Jewish population of southern Morocco is a specific part not only of the local patriotism of individual homesteads, but also of the wider Berber identity [11].

*1.3. About the Modern 3D Documentation in Cultural Heritage*

Unfortunately, disappearing monuments are a reality. There are various reasons for the destruction of monuments: Civilization development, the construction boom, wars, terrorism, weather, and disasters, etc. Not all monuments can be saved, not all are so valuable that society can find the financial resources to save and restore them. It is therefore necessary to at least document the disappearing monuments. This is the subject of the whole scientific field.

Accurate documentation of historic buildings is a long-term matter. Historically, drawing was used, and later, classical geodetic technologies and photographs. In the 20th Century, mainly terrestrial or aerial stereophotogrammetry and terrestrial geodetical instruments like theodolites were used for mapping and spatial documentation of historical object or construction [12]. By the end of century, new technologies for documentation and mapping were integrated into cultural heritage, based on advances in electronics development that have affected most areas of human activity [13]. Very-high-resolution satellite images (VHR) were intensively used with meter or sub-meter resolution, laser scanners of various construction have been developed as aerial laser scanners (ALS), terrestrial laser scanners (TLS), or mobile laser scanners (MLS) [14]. Photogrammetry was fully digitized with the transition to the new millennium and procedures based on image correlation were implemented into documentation process. These are known under different abbreviations such as IBMR (Image-Based Modeling and Rendering) or SfM (Structure from Motion). Over time, data from different technologies began to join and process together [15,16].

Electronization, digitization, and other modern methods of computer technology and visualization, such as virtual reality (VR), computer vision [17,18], and BIM (Building Information Modeling) [18,19], are increasingly entering the care of monuments. Classic documentation becomes a multisensory technology [15]. RPAS (Remotely Piloted Aircraft System) has often been used to document historic buildings in the last decade [20,21]. Documentation of historic buildings depends on their origin. Those found in North Africa and the Middle East are specific to the material, construction, and natural conditions used [22].

## 2. The University Project "Morocco" and Project Aims

*2.1. About the Project*

The project, "Jewish Traces in Southern Morocco," was initiated by employees and students of the Department of the Middle East at the Faculty of Arts, Charles University (FF UK), in cooperation with the Department of Geomatics at the Faculty of Civil Engineering (FCE CTU) and the Institute of Monument Care at the Faculty of Architecture (FA CTU). Together, they formed a multidisciplinary team consisting of experts in synagogue architecture, Judaism, North African culture and dialects, folk architecture, and monument documentation.

Since 2018, this multidisciplinary team has been compiling a detailed map of historic Jewish homesteads in southern Morocco based on studying secondary literature and literary, photographic, and eyewitness accounts. The team has already undertaken three expeditions to this region, during which it began to systematically research the current state of selected localities, to review documentation, and to analyze Jewish monuments and their role in the thinking of the Muslim inhabitants of individual municipalities.

At the beginning of February 2020, a third expedition to southern Morocco took place in order to research and document the disappearing Jewish monuments (Figure 3).

The Department of Geomatics of the Czech Technical University, Faculty of Civil Engineering, which has the appropriate technology, was invited to do so. As this was always a low-cost expedition, previously only classical methods of mapping objects using digital rangefinders and hand-drawn plans were used. This is generally slow and inaccurate for irregular and often partially damaged objects.

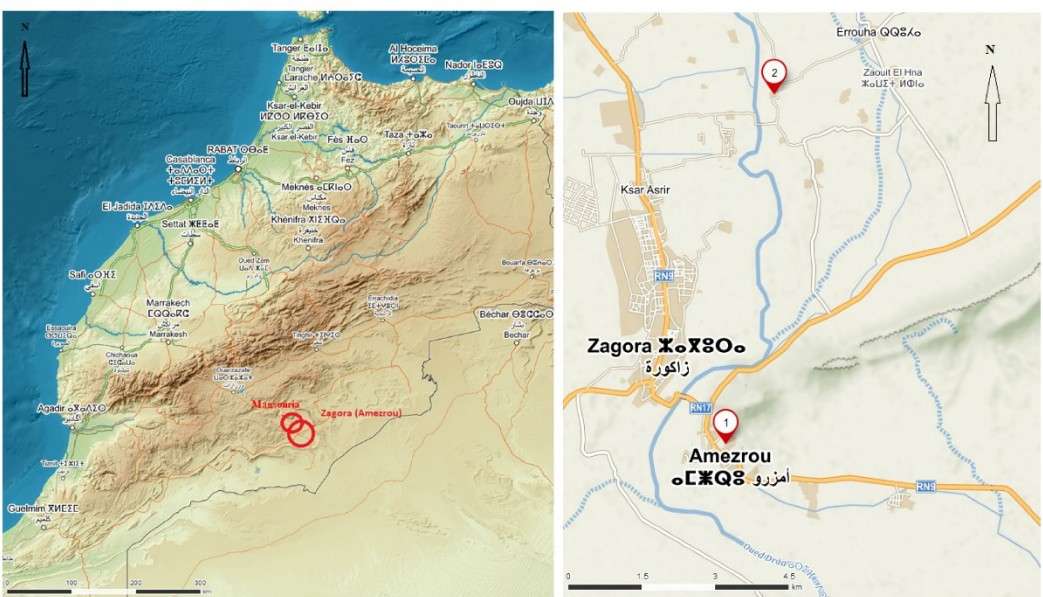

**Figure 3.** Map of Morocco with the location of both studied objects.

### 2.2. The Project Aims

Based on the cooperation of the above-mentioned university professional workplaces, the aim was to perform and test modern, fast, and accurate 3D documentation of objects of interest. Today's documentation of historic buildings presupposes the use of modern technologies that will significantly speed up the documentation process [23–25].

The main aims of this expedition were:

(a) To define accuracy of new modern easy to use and transportable laser scanners,
(b) To determine the possibility of economical and effective technology for documentation of this specific cultural heritage; the target group: Cooperation with historians from the Faculty of Philosophy, the Charles University in Prague and Moroccan historians from the state monument institution (CERKAS),
(c) On case projects, to research usability of PLS, TLS, and close-range photogrammetry,
(d) To show the quickness and accuracy of results on some case projects, which were very difficult to measure with a traditional approach,
(e) To give a recommendation for future work in this area.

## 3. Instruments Used and Data Capturing

### 3.1. Selection of Instruments

Low budgets and transportation problems have forced the use of miniaturized instruments. For this reason, it was not possible to use classic total geodetic stations or larger laser scanners on tripods. Therefore, the most modern methods of contemporary geomatics were employed using portable measuring instruments.

In Morocco, there is also a problem with the authorization of drones for photogrammetric documentation; drones or in general RPAS (Remotely Piloted Aerial System) were very suitable for some hardly accessible objects or areas in this project [26]. Of course, there are more possible technologies. Remote sensing or aerial images can be used instead of the often-problematic use of RPAS. However, the problem here is that most of the objects of interest are roofed and the aerial photographs do not give much information, only about the shape of the entire building system from the outside, but not from the inside.

Furthermore, easily portable devices such as 3D scanners and both stop-and-go and mobile systems are ideal for basic documentation [27]. Modern close-range photogrammetry can also be used. It turns out that it is ideal to use more technologies,

because each documentation project contains several types of objects, and different procedures of their measurement are suitable [28].

The constructions we documented during the expedition were visible from the outside like monotonous walls of unfired clay blocks. The aim was to document the interior spaces especially, which were extensive, dark, and narrow. These are conditions that are not suitable for the use of close-range photogrammetry. Therefore, PLS technology was used.

### 3.2. Instruments Used

Modern miniaturized and easily transportable laser scanners were used (Figure 4). The Leica BLK360 miniaturized laser scanner with a small carbon tripod and built-in camera was a good choice for the precise documentation of valuable objects. Nevertheless, even with this device, the documentation takes a relatively long time (up to several hours of measurement is typical for an object with dozens of scanner positions). BLK360 has a precision 4 mm on 10 m and a scan rate to 360,000 points per second. One scan takes (with the capturing of photos) about six minutes.

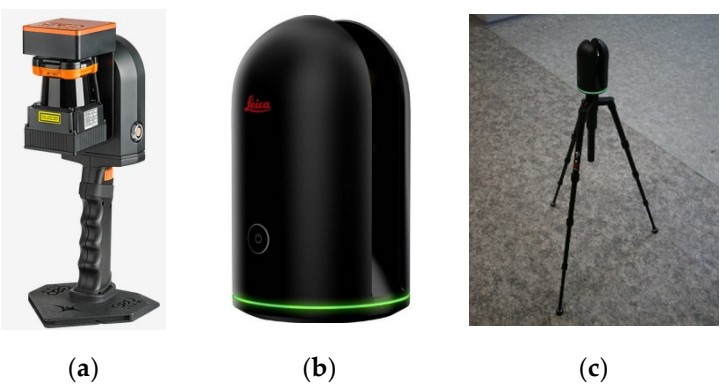

(**a**)              (**b**)              (**c**)

**Figure 4.** (**a**) Laser mobile scanner ZEB-REVO, (**b**,**c**) laser scanner BLK360.

Due to the tense itinerary and the often-complex agreement with the local citizens, the ZEB-REVO mobile personal laser scanner (PLS) was used successfully. It is a laser handheld scanner equipped with an inertial unit, which uses SLAM (*S*imultaneous Localization and Mapping*)* technology to join data and mapping [29–31]. Although the ZEB-REVO does not have the millimeter accuracy that conventional laser scanners have, it excels in speed and mobility of measurements. For buildings that are made of unfired bricks, partially damaged, very irregular, and full of corridors with poor lighting, it was an excellent choice. A completely regular covered building with an area of more than one hectare could be documented in about one hour. The device is slowly carried by the operator, starting and ending at the same place. The system keeps navigation for up to 50 min without further intervention or information. Incremental errors are divided in the point cloud during postprocessing because the measuring starts and ends at the same place. The difference found is divided into a whole trajectory. Data processing is fully automatic, with accuracy in this case sufficient, reaching 1–3 cm in a position with a range of up to 30 m (scan rate: 43 thousand points per second). The amount of data from the two above-mentioned scanners is also significantly different. A typical single scan from the BLK360 device has 600 MB; data from a 40-min walk through an object with the ZEB-REVO has about 300 MB. Sixty-eight scans with the BLK360 scanner and 21 scans with the ZEB-REVO scanner were performed on 10 objects in five days.

A standard Canon 450 SLR or simple a smartphone was used for close-range photogrammetry in some cases (details, rooms, artifacts).

### 3.3. Accuracy Testing of BLK360 and ZEB-REVO Scanners

The devices usability and their actual accuracy were first tested in the basement of the Faculty of Civil Engineering, Czech Technical University in Prague before departure to the Morocco expedition (Figure 5). For the purpose an objective analysis of the tested device accuracy, the basement space was mapped using manual measurements, then the Leica TCR3 total station was used, and three types of laser scanners followed. To compare the laser data, the most accurate laser phase scanner Surphaser with an accuracy of 0.6 mm/10 m was used as a reference model. These data were compared to results derived from a small BLK360 scanner and a ZEB-REVO mobile personal scanner (Figure 6).

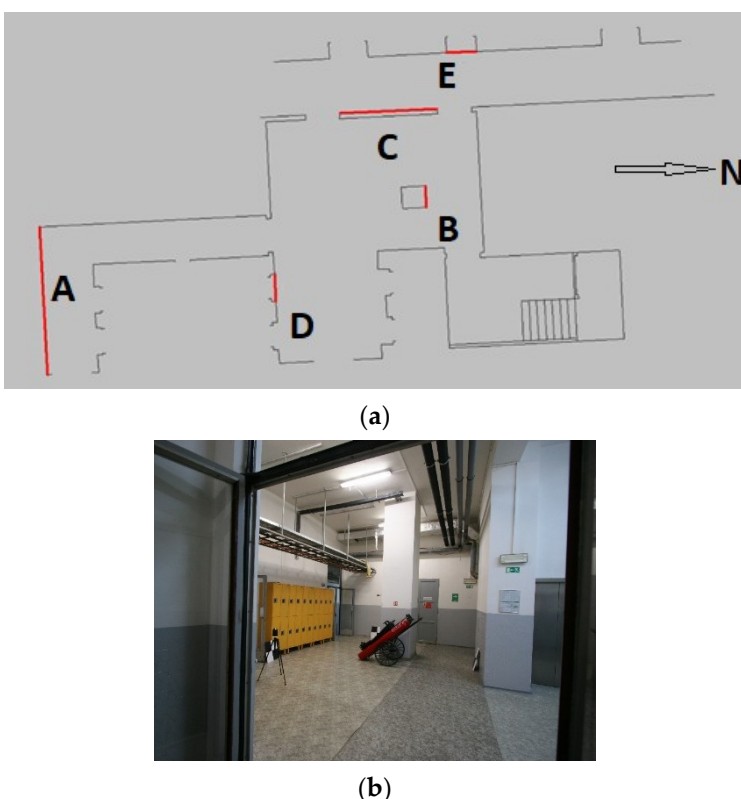

(**a**)

(**b**)

**Figure 5.** (**a**) Floor plan and analyzed distances in the faculty basement, (**b**) a view of the faculty basement part.

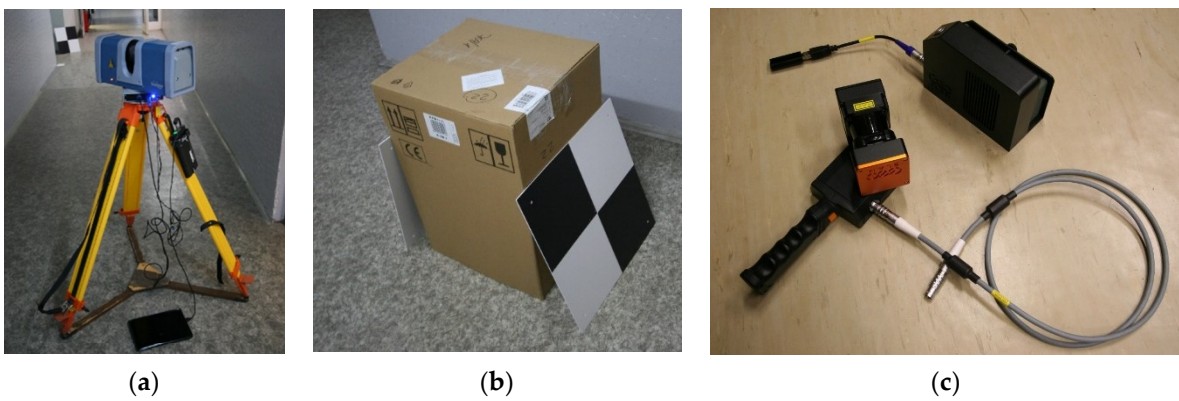

(**a**)                (**b**)                (**c**)

**Figure 6.** (**a**) Precise laser scanner Surphaser 25 HSX, (**b**) used targets for point clouds alignment, and (**c**) ZEB-REVO personal mobile laser scanner.

The derived results were processed in the form of a table showing selected measured distances in the documented space (Table 1).

**Table 1.** Results of selected measured distances (A, B, C, D, E) using different instruments.

| Instrument | A [m] | B [m] | C [m] | D [m] | E [m] |
|---|---|---|---|---|---|
| Leica Disto A5 | 5.761 | 0.835 | 3.771 | 1.048 | 1.166 |
| Leica TCR 307 | 5.758 | 0.835 | 3.758 | 1.056 | 1.174 |
| Surphaser 25 HSX | 5.761 | 0.836 | 3.781 | 1.049 | 1.169 |
| Leica BLK 360 | 5.767 | 0.833 | 3.775 | 1.043 | 1.164 |
| ZEB-REVO | 5.785 | 0.850 | 3.779 | 1.028 | 1.174 |

Point clouds from laser scanners were compared to each other using a CloudCompare software.

The comparison function Compute cloud/cloud distances was used to calculate distances between identical groups of cloud points. The function provides statistical indicators comparison in the form of arithmetic mean and standard deviation of distances between identical points; clouds can be colored according to the calculated deviations to visualize the comparison results.

The arithmetic mean of the distances differences between the points derived from the Surphaser 25 HSX and Leica BLK 360 scanners was 6.35 mm with a standard deviation of 10.38 mm.

The arithmetic mean of the differences in distances between points measured by the Surphaser 25 HSX and ZEB-REVO scanners was 10.12 mm with a standard deviation of 12.76 mm

This test showed the applicability of both types of laser scanners for the documentation of historic buildings during the planned expedition.

A comparison of the easy-to-transport instruments used is shown in the following table (Table 2).

**Table 2.** Comparison of accuracy and economy of the process (* including software, ** without software price—we use the university license, which is only 0.6 K €).

| Instrument | Accuracy [mm]/on 10 m | Range [m] | Speed of Scanning [points]/sec | Approx. Data Volume for the Same Object Scanned [GB] | Time [minutes] | Instrument Approx. Price [K €] |
|---|---|---|---|---|---|---|
| BLK360 | 4 | 30 | 360,000 | 0.6/1 scan | 6 per scan/middle resolution | 30 * |
| ZEB-REVO | 10–20 | 30 | 48,000 | 0.1/per five minutes | slow walking speed | 30 * |
| Photogrammetry IBMR (a DSLR, here Canon 450) | 5–20/depend on camera type and on distance | 10 | Up to 30 photos/minute | 0.04/per photo | Up to 30 photos/minute | 1 ** |
| iPhone 7 camera | 5–50 depend on distance | 5 | Up to 30 photos/minute | 0.04/per photo | Up to 30 photos/minute | 1 ** |

### 4. Methodology

The aim of the documentation was plans and cross-sections of buildings, especially roofed *mellahs*, as well as 3D documentation of buildings or special spaces like synagogues. The basis was therefore always spatial measurement. From the point of view of methodology, all technologies (MLS, TLS, and IBMR close-range photogrammetry) were used (Figure 7).

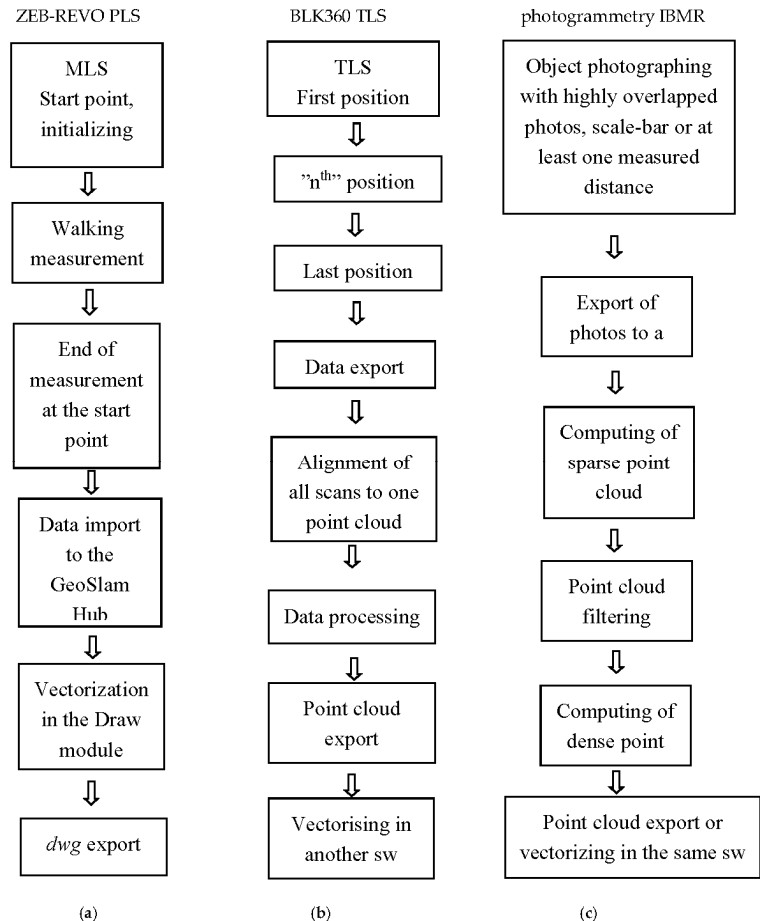

**Figure 7.** Schematic workflow for (**a**) personal laser scanner (PLS) ZEB-REVO, (**b**) TLS BLK360, and (**c**) for Image-Based Modeling and Rendering (IBMR) photogrammetry.

The hand-held PLS ZEB-REVO was used as the basic device and photogrammetry and TLS were used as complementary technologies. It must be said that each has its advantages and disadvantages and their use is different for different objects.

The PLS is speedy, mobile, and ideal for mapping larger built-up areas; in the case of the ZEB-REVO there is no texture, because the ZEB-REVO scanner used here does not have a camera and only creates a non-textured point cloud. It has an accuracy within 1 cm, which was sufficient for the purposes of expedition and given objects, and it means often considerably decayed *mellahs*. TLS is more laborious, while with the BLK360 there is texture and higher accuracy (Figure 8); the disadvantage is the low speed of documentation compared to PLS. Photogrammetric documentation is currently at a very high and fast level in the form of SfM or else IBMR, which produces, similarly to laser scanners, a point cloud [32,33]. In the case of photogrammetric technology, the point cloud is always textured. It can be said that even photography is a certain form of scanning but using a matrix of detectors and usually in an irregular sequence. It is necessary to add a scale bar or measure at least one distance on the object, because photogrammetry does not work with metric units, but in pixels and lines. The disadvantage of close photogrammetry during this expedition was the strong Sun, common in Morocco. It makes strong shadows, which create an incorrect texture during processing. Even so, documented and mapped objects were often dark interiors, where a flash or special lighting is required. It was not possible to photograph long, irregular, and dark corridors and create a point cloud from the photographs because of both lack of time and appropriate equipment like studio lighting. For this reason, only some objects were documented photogrammetrically like special building details, artifacts etc.

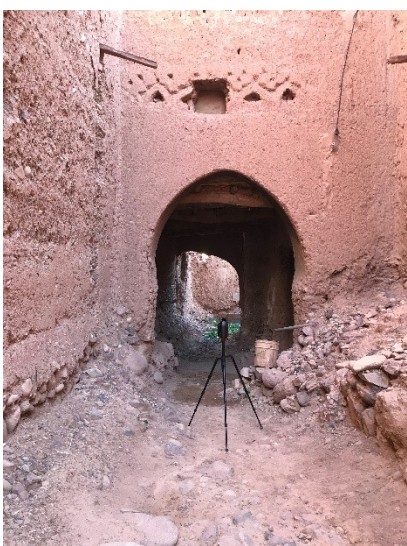

**Figure 8.** Laser scanner BLK360 during the measurement inside the *mellah*.

## 5. Data Processing

In general, the use of laser scanning in the documentation of monuments has become very widespread, but it has also brought a large increase in data [34,35]. Over 48 GB of data were acquired in five days, including photographic ones. The schematic procedure is shown in Figure 7. The BLK360 is a highly sophisticated device aimed at maximum simplicity. However, if you take dozens of scans, they cannot be processed online on the tablet. The data are further processed almost fully automatically in the Leica Cyclone REGISTER 360 software (BLK Edition) without much possibility to influence the process. At best, the scans are automatically joined based on the correlation and a report is created.

However, it is necessary to make a large overlap of scans, which can only be based on experience. For simple objects, the scans join well automatically based on overlap, typically more than 50% for more complex ones, sets of joined scans are created, which must be joined into one manually via tie points. However, the result is a relatively high-quality 3D model with texture. Any vectorization must be done in other software like REVIT for example; BLK360 is primarily intended for quality 3D models.

The situation is different for the ZEB-REVO—this device is intended to create plans or as an input for BIM. The device automatically saves the data after the measurement, which is simply transferred via cable and USB to a computer and uploaded directly in compressed form to the GeoSLAM HUB software. A 3D model is created here; this can be transferred by mouse-clicking to another module, where a floor plan and two side views are created. However, the simplest ZEB-REVO model does not have a camera and you cannot see what is being scanned; this can have processing consequences. With long and complicated object documentation, such as a narrow high tower or a very large object, the partial bifurcation of the resulting model can occur due to the accumulation of errors from the IMU. In general, however, automatic processing can affect parameters; their setup is a matter of experience.

When documenting an object, a vector model is usually required as a result, most often as a floor plan and cross-sections. The Geoslam software serves as a "Draw" module, which allows the vectorization of the created orthogonal projections of the point cloud to floor plan and to defined point cloud cross-sections. Next, a vectorization procedure can be applied, which works semi-automatically. It is necessary to define the amount of vectorized information by data thresholding before starting automatic vectorization. This procedure finds vectors automatically, but for complex and irregular objects, especially historical ones, it is necessary to edit the result, often fundamentally. Even so, it is possible to create a floor plan of the building in a few hours.

From the point of view of close-range photogrammetry in the IBMR (SfM) form, the situation is known; if enough of overlapping images is taken, a 3D textured model in form of point cloud and textured mesh can be created based on image correlation. The number of photos taken depends on which camera was used, mainly on camera-lens, the size and complexity of the documented object and the required accuracy and detail of the output, and on the possibilities of access to the object. You can use several types of professional software, such as Metashape, Zephyr, etc. [36,37]. Many authors have compared the performance and results of various software for close-range photogrammetry [38].

There is also open-source software like Mic-Mac, Bundler, Photosynth, 123catch, etc., which can be successfully used as a low-cost photogrammetric solution; in this case, only both suitable digital camera and a computer are necessary.

If hundreds of pictures were taken, it is necessary to use an efficient workstation and the process can take hours. With a high overlapping between the images and ensuring that all important areas of the object are visible in at least three images, a dense point cloud with the useful 3D information will be produced. A proper strategy during capturing the image data will guarantee a geometrically accurate result without missing parts. Vectorization can be made in additional software.

## 6. Case Projects

In this article, two case projects are described as examples of using modern technology for 3D historical object documentation.

The portable and easy-to-use laser scanners Leica BLK360 and ZEB-REVO were mainly used to document the objects. It was more convenient to document some objects using the BLK360, as this scanner also includes a digital camera, which the model can be textured with. It was certainly more suitable to document larger objects made of unfired clay, where the aim was to find out the exact structure, with the ZEB-REVO mobile scanner.

### 6.1. Synagogue in Amezrou

The synagogue in the city of Amezrou (near the Zagora, Figures 9 and 10) is a unique monument, now under reconstruction. It is a unique triangular synagogue (approx. 7 × 7 × 10 m³) located in the *mellah*, which was a part of the old town. The synagogue was documented and mapped with the help of both laser scanners (BLK360 and ZEB-REVO) (Figures 11 and 12). The main aim for historians and architects was to get a precise shape and orientation of this sacral place. The demonstration of new technologies was also intended for local monument care specialists, who gave us a short and rare opportunity to see the place.

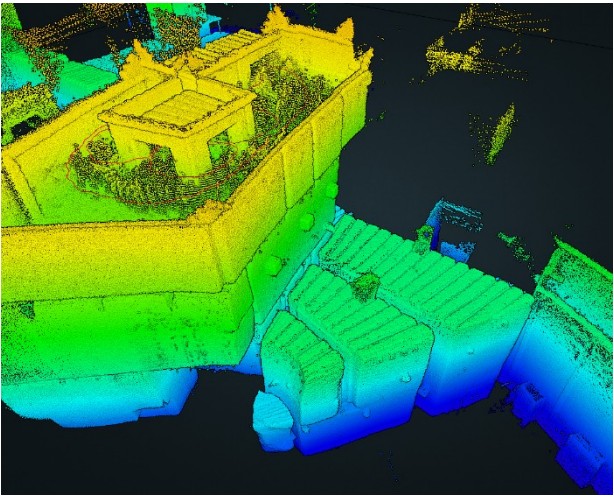

**Figure 9.** Graphical view on the captured point cloud coloured by hight.

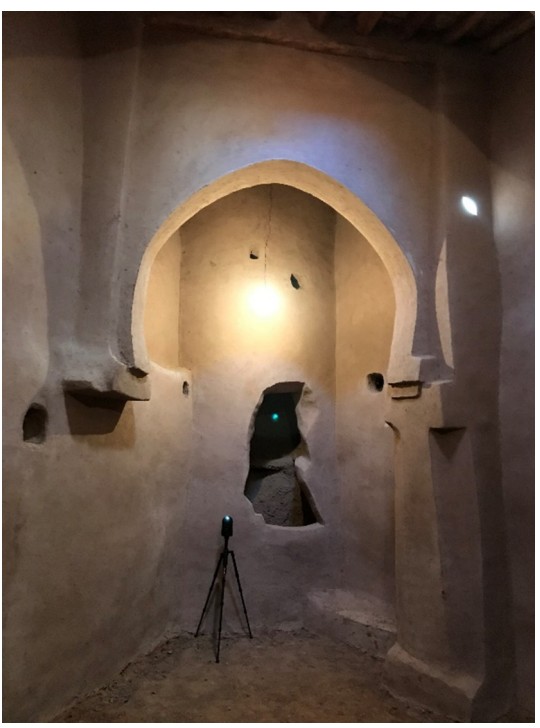

**Figure 10.** Synagogue in Amezrou (inside BLK360 laser scanner).

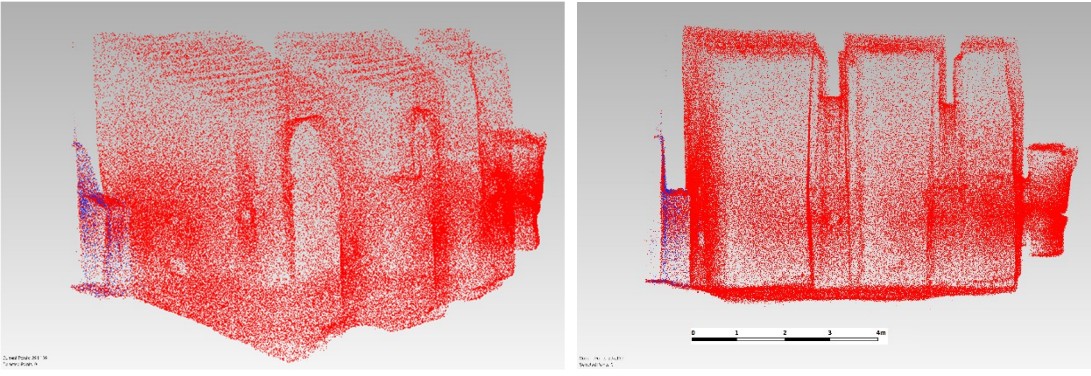

**Figure 11.** Point cloud from ZEB-REVO (a camera for texturing is not included in this model).

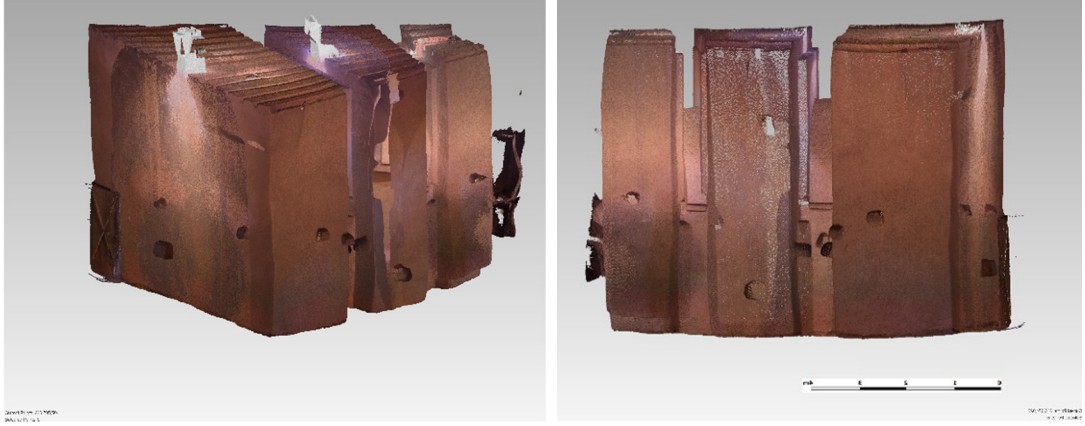

**Figure 12.** Textured point cloud from BLK360.

By using of ZEB-REVO, the synagogue was measured in three minutes only by walking through synagogue. There are no other characteristics that would indicate the

quality of the measurement. Processing is completely automatic. The only option is to use control points or compare with a model obtained from a more accurate device.

Photogrammetry was made on this place as a technological experiment only. Due to insufficient lighting for high-quality close-range photogrammeters, images were taken only by a mobile phone (iPhone 7:12 MPix camera with 4 mm focal length, Sony Exmor RS 1.22 µm pixel size). One hundred and forty-one overlapped images were taken in 15 min. A model created from the photographs was not sufficient and missing parts occurred because 10% of all photos was not possible to orient due the shadows and low overlap (Figure 13). The factors that determine the quality of the produced 3D model are mainly lens distortion, low-resolution in texture mapping, low lighting leading to the image blurring, and colour blurring on the image border. Of course, a smartphone is not ideal equipment and it was not the goal to use precise photogrammetry with this camera (Table 3). However, the camera quality in smartphones is constantly improving and with the right photographing process, the results may be satisfactory for smaller subjects. In this case, it was just a test, which shows that low-cost close-range photogrammetry without precise preparation can get insufficient results (Figure 13), but in other cases for documentation of small artefacts for example, it gives results of sufficient quality with sub-millimeter GSD (ground sample distance). The results from all technologies are shown in Figures 14 and 15.

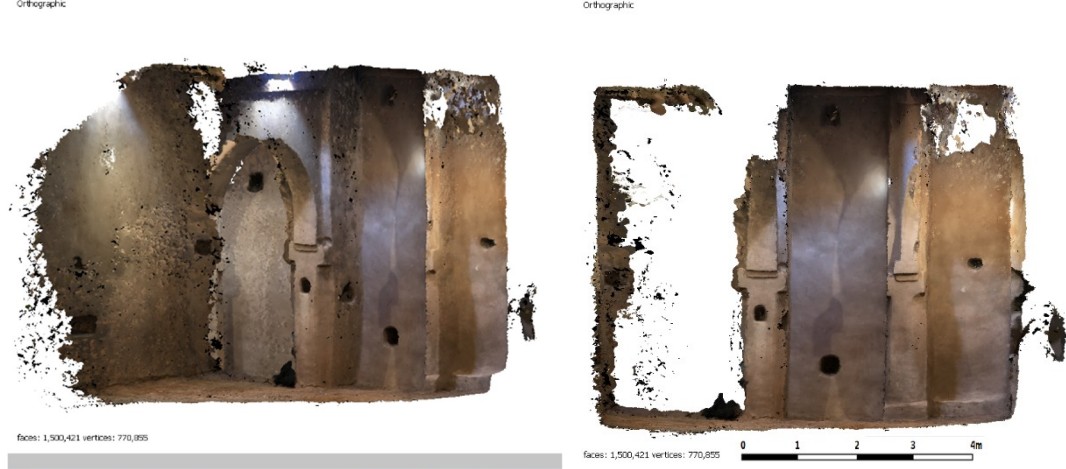

**Figure 13.** Point cloud created using Metashape software; a test with mobile phone camera (iPhone 7)—it was not a goal to make a photogrammetric model because we had other devices and there was no lighting equipment or a tripod at our disposal.

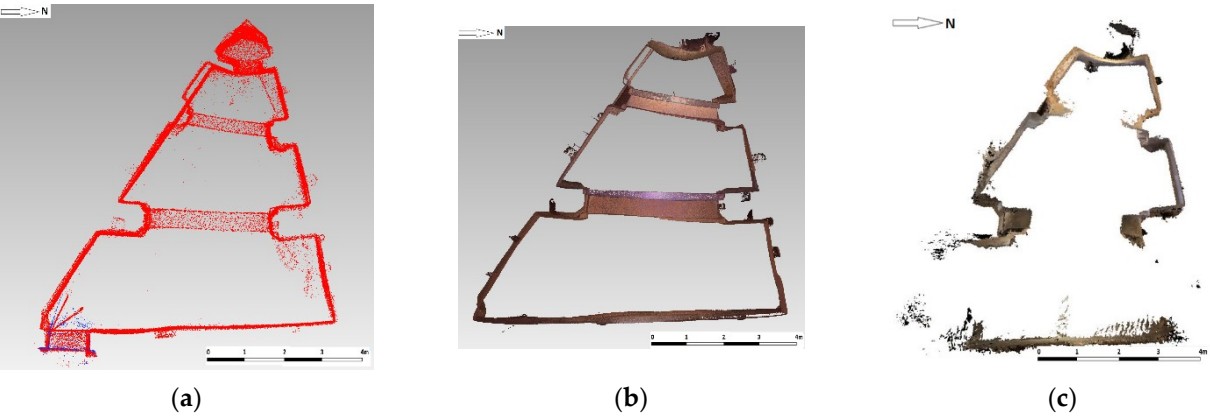

| (**a**) | (**b**) | (**c**) |

**Figure 14.** Ground schemes from ZEB-REVO (**a**), BLK360 (**b**), and iPhone (**c**); you can see that with the handheld scanner, the spaces behind the wall (behind the tip of the triangle) can also be documented. The model from photos, taken by an iPhone 7, was not very good due to bad lighting, inappropriate texture, combined with a short time for taking photos; however it is not a typical result from close-range photogrammetry, that can create high-quality 3D models..

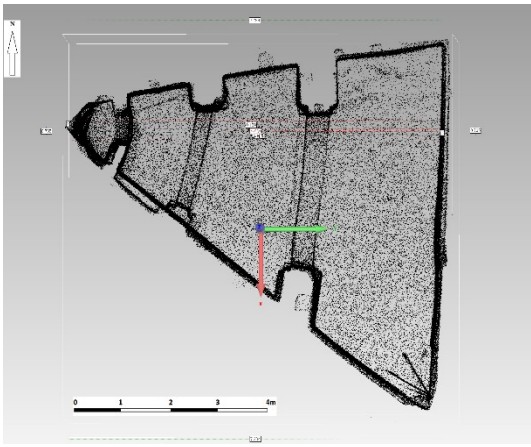

**Figure 15.** The ground-plan of the synagogue from the point cloud (ZEB-REVO).

**Table 3.** Comparison of the instruments used for the synagogue in the Amezrou.

| Instrument | Measurement Time [minutes] | Data Processing (to the Point Cloud) [minutes] | Point Cloud [points] | Accuracy [mm] |
|---|---|---|---|---|
| BLK360 | 30 (four positions) | 15 (data transferring) + 30 (data processing) | 220,000,000 | 4–5 |
| ZEB-REVO | 3 | 5 (data transferring) + 10 (data processing) | 300,000 | 10–20 |
| iPhone 7 (IBMR) | 15 | 5 (data transferring) + 20 (data processing, Metashape) | 70,000,000 | 5–50 (variable, depend on the data-noise and model scaling) |

The orientation of the synagogue was measured using the iPhone build-in compass. This is for information only, showing how the synagogue is oriented. No correction was applied because the magnetic declination is here only 0.66 degrees. In the Jewish sense, the altar is a cabinet built into the front wall of the synagogue and it contains Torah scrolls. The altar is directly to the east, and in the model, it is a built-in cabinet located behind the tip of the triangle. A precise orientation (east–west) was measured also in other documented synagogues (Figure 16).

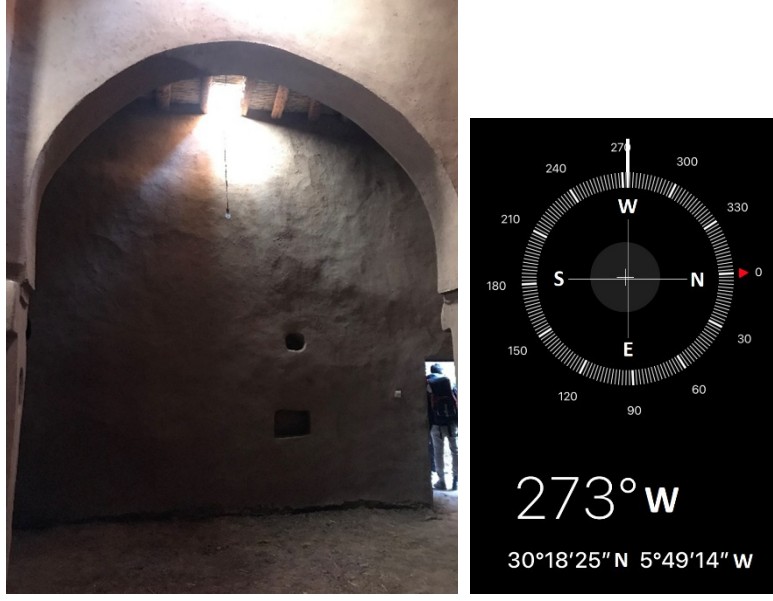

**Figure 16.** Orientation of the synagogue (the compass from the iPhone).

### 6.2. Mansouria (Zagora)

Near the city of Zagora there is a small, well-preserved village called Mansouria (Figure 17); it is a closed type with an internal structure and with a *mellah*. This village was documented from the outside and inside using the ZEB-REVO scanner (Figure 18a,b).

Measuring with the ZEB-REVO is easy. After activating the device, it is necessary to go slowly through the documented object with the scanner; it is not possible to make sharp movements or changes of direction, to stay standing, and to map homogeneous areas without existing spatial structures. It starts and ends at the same place, a typical scanning time is suitable for about 20–30 min, but you can do a scanning for up to 50 min. The maximum time in the project was over 40 min. For longer scans, there may be a problem with bifurcation of the model due to the accumulation of IMU and SLAM errors.

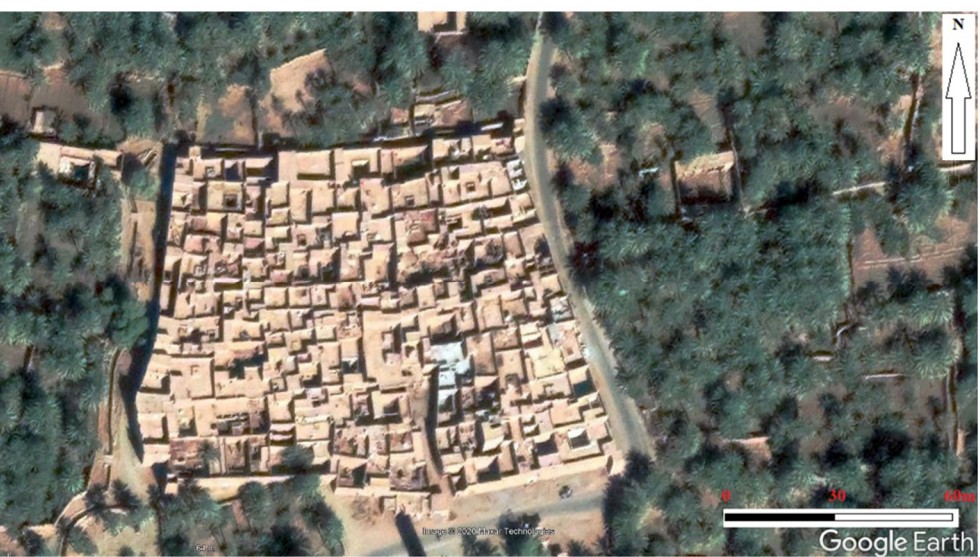

**Figure 17.** The village of Mansouria from Google Earth.

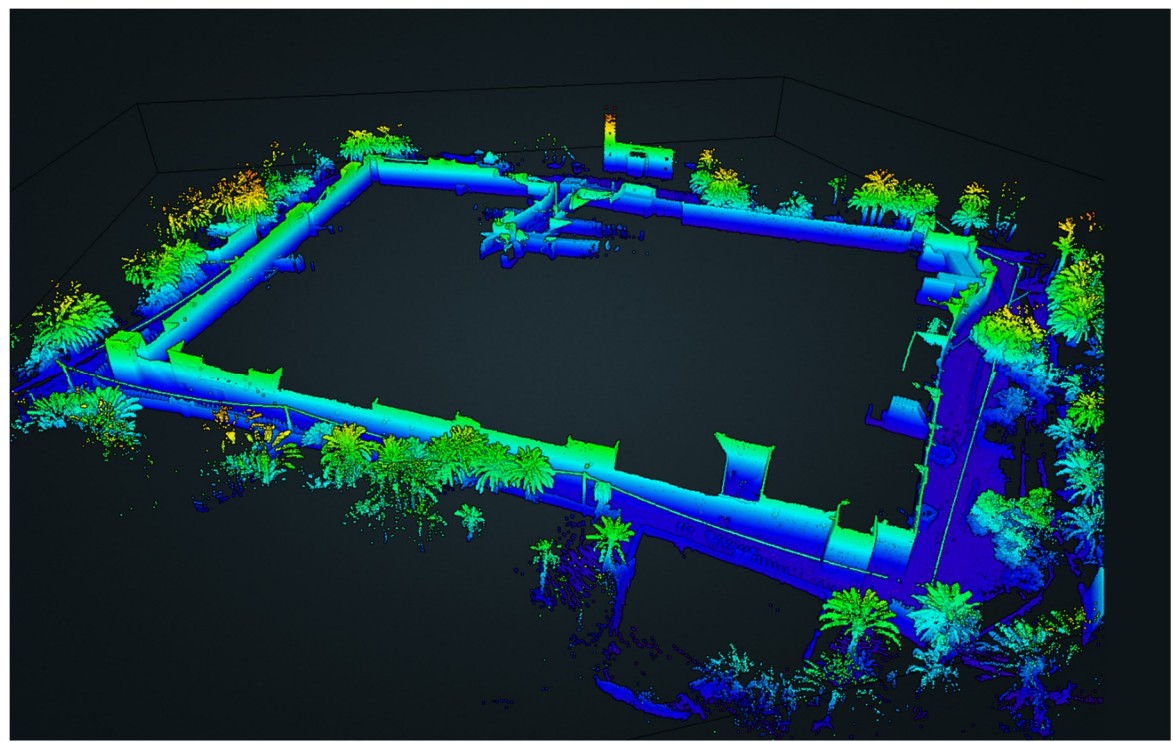

(**a**)

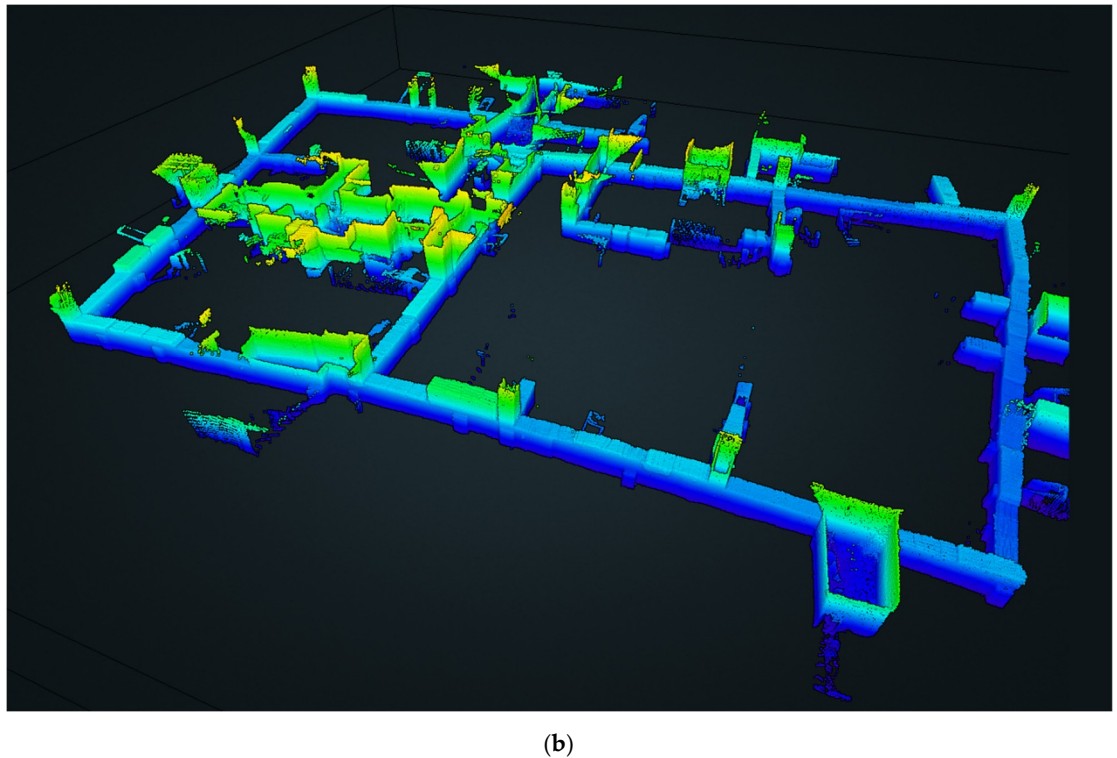

(**b**)

**Figure 18.** The village Mansouria. Outside view (**a**) and inside view (**b**) on a 3D model created using ZEB-REVO in the GeoSlam HUB.

When scanning is completed, the data are saved automatically. They are transferred to a computer using a flash drive, where the calculation of the model is performed in the GeoSlam HUB completely automatically. The model can then be semi-automatically vectorized in the Draw module (Figure 19a,b). This works well for simple and regular objects; significant editing work was required for objects that were documented during the 2020 Morocco expedition. The results are automatically saved in CAD format (Figure 19).

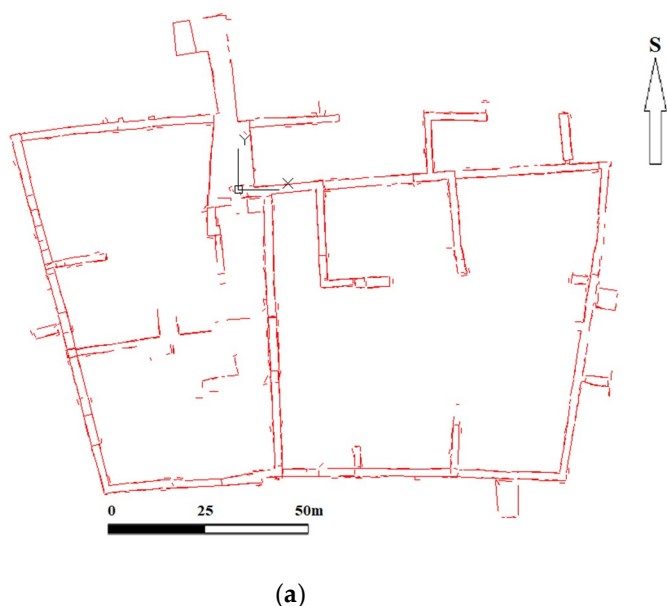

(**a**)

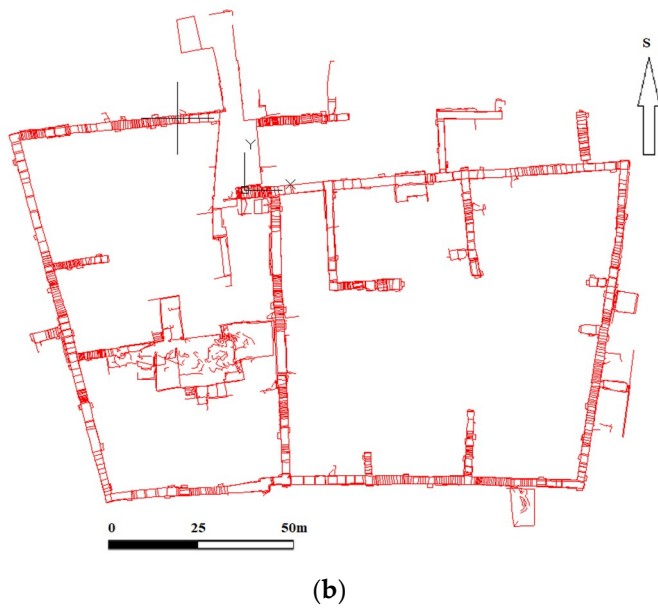

(**b**)

**Figure 19.** The village of Mansouria. (**a**) An original plan after an automated vectorization from GeoSlam Draw, Mansouria (Zagora); (**b**) an edited vector plan from GeoSlam Draw (approximately four hours of editing of the original automated vectorizing).

## 7. Results and Discussion

### 7.1. Laser Scanning

#### 7.1.1. Amezrou Synagogue

A small object (interior of synagogue in Amezrou) was documented by both scanners, using photogrammetry too, so it was possible to define the mutual accuracy and work efficiency (Table 3).

By working with BLK360, only four scanner position were used, located approximately in the corners. The measurement took approximately 30 min. All four scans were processed after expedition in Cyclone 360 Register software. Joining of collected scans was a fully automated process based on correlation as "cloud-to-cloud" process. Finally, a report was prepared, which showed accuracy characteristics (Table 4). Six links were found between four scanner stations.

**Table 4.** Results from the BLK360 laser scanner, the synagogue in the Amezrou.

| Link Nr. | Combination | Overlap [%] | Abs. Mean Error [m] |
| --- | --- | --- | --- |
| Link 1 | 1–2 | 63 | 0.004 |
| Link 2 | 1–3 | 65 | 0.003 |
| Link 3 | 1–4 | 95 | 0.003 |
| Link 4 | 2–3 | 93 | 0.003 |
| Link 5 | 2–4 | 63 | 0.004 |
| Link 6 | 3–4 | 63 | 0.003 |

The final joined point cloud, created from BLK360, was set as a reference measurement because this scanner produces relatively precise measurement on short distances, in this case, 3–4 millimetres in measured distances.

Measurement with the ZEB-REVO scanner was very fast and simple (Figure 20). Since no geodetic control points were used to test the accuracy, the joined point cloud from the BLK360 scanner, which has an order of magnitude higher accuracy, was used to compare the accuracy of this new PLS device.

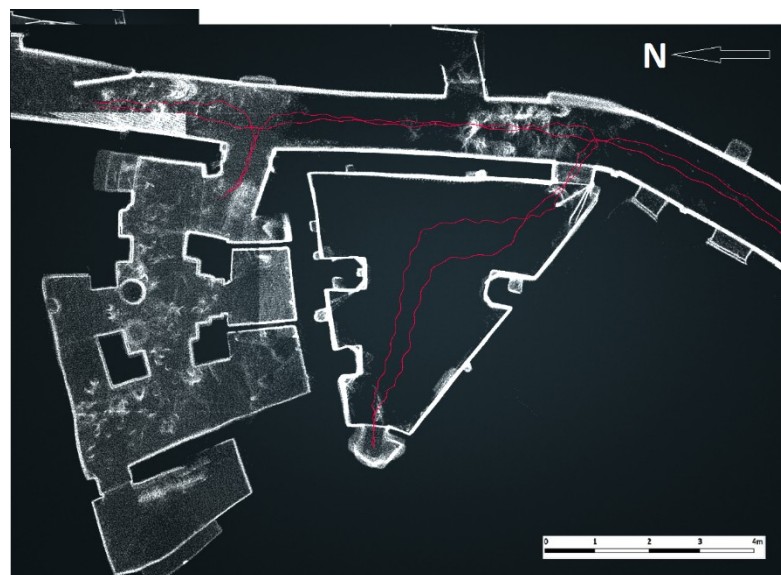

**Figure 20.** Graphical output from the ZEB-REVO PLS: The ground plan of structures inside the *mellah* (in red colour the trajectory during measuring). This output can be produced in dozens of minutes.

Close-range photogrammetry was used only in addition. Due to lack of time and poor lighting condition, only a mobile phone camera was used.

All the data were processed at the university post-expedition. From laser scanners, point clouds were computed and exported to the CloudCompare software (Figure 21) As a reference point cloud, the BLK360 measurement were selected, because it was complex and most accurate from all three methods. Differences after automatically joining of four measured point clouds reaches only 3 mm and 4 mm, respectively, with average overlap more than 80% based on the computational report (Table 4).

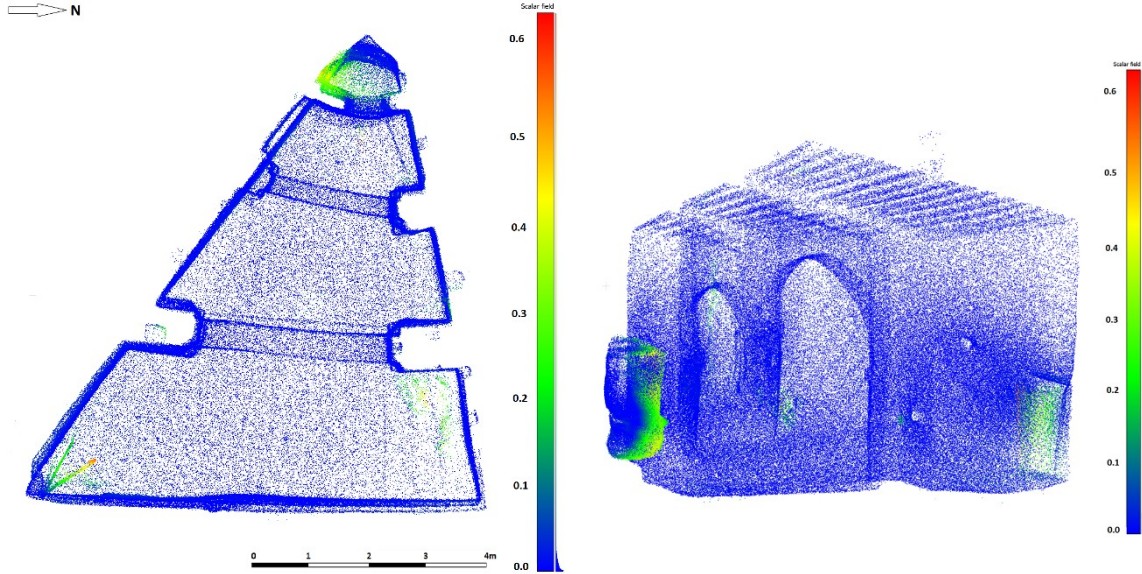

**Figure 21.** Comparison of joined point clouds from both laser scanners in CloudCompare (CC) software.

This final point cloud from BLK360 was compared with the point cloud created by ZEB-REVO PLS. It is visible here (Figure 21) that absolute differences between both point clouds are under 2 cm, typically 1 cm only. A visible difference is only on the left

(somebody has opened the doors), on the top (the tips of the triangle are not visible in BLK360 point cloud), and right (our relocated backpacks).

Further accuracy comparisons were made on cross-sections (Figure 22). The variance of data noise and point density manifested itself here.

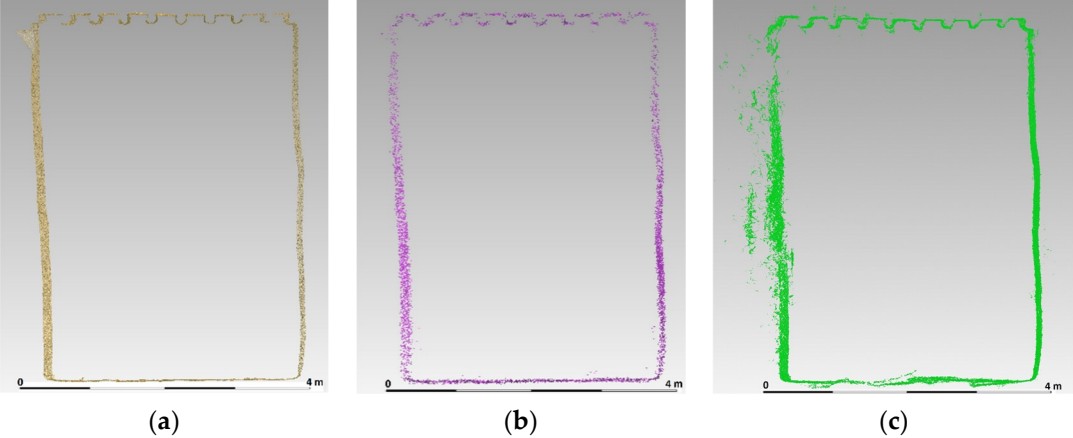

| (a) | (b) | (c) |

**Figure 22.** Comparison of the cross-section from (**a**) BLK360, (**b**) Zeb-REVO, and (**c**) close-range photogrammetry (iPhone camera). At the top: A ceiling made of beams and reeds.

It can be seen in Figures 22 and 23 that due to unsuitable conditions, the point cloud from close-range photogrammetry contains considerable noise and errors (green dots, variance up to several cm). The point cloud from the BLK360 scanner is accurate and contains very little data noise (variance is only within 2–5 mm). The data from the ZEB-REVO scanner are sparse when compared to the two named outputs and has significant variance in points, typically 1–2 cm. It can also be seen that photogrammetry data are systematically shifted by a value of approximately 1 cm; this is probably due to the material structure and its reflectivity, and by the photogrammetrical model deformation [39].

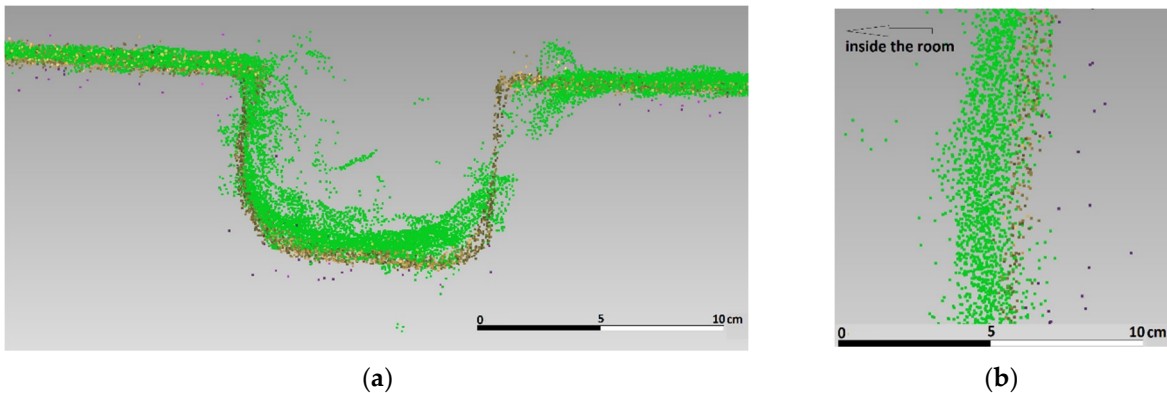

| (a) | (b) |

**Figure 23.** Comparison of joined point clouds from BLK360 (brown color points), Zeb-REVO (purple color points), and close-range photogrammetry (iPhone camera, green color points); (**a**) wooden beam on the ceiling, (**b**) vertical wall in the middle of the synagogue at 1.5 m height.

In this case, the geometrical accuracy comparison of scans from the BLK360 and ZEB-REVO scanners can only be done as merging of scans from the same object and as a deviation comparison. The BLK360 has certainly better geometric accuracy—i.e., the point cloud from the BLK360 was taken as a reference (there are no special control points measured, for example, precise geodetical total station). The comparison was made using a CloudCompare software by comparing both point clouds together. As shown on Figure 21, the result—comparing both point clouds—is accurate enough and reaches 1–2 cm on

stable surfaces. Some small parts of the point cloud from the BLK360 scanner were missing compared to the point cloud from the ZEB-REVO scanner. This is due to inaccessible spaces for the BLK360 or an accidental opening of door during measurement.

### 7.1.2. The Village Mansouria

Dozens of other objects of various sizes were also documented. Some of the objects were the size of an entire roofed village—the ZEB-REVO mobile laser scanner, with its advantageous high speed, was used here. Smaller sections were documented by the BLK360 scanner (better resolution and photographic texture), and close-range photogrammetry was used for rooms, artifacts, archaeological finds, or small objects. With such extensive documentation and low expedition costs, however, it is not possible to use only one technology.

For a small building, all three technologies can be used in general, but for larger, irregular buildings with many narrow-roofed corridors, it is quite clear that ZEB-REVO or similar equipment based on SLAM technology is unmatched in terms of economy and speed. It is certain that in enclosed dark spaces, which were documented during the expedition in a short time, it was not possible to define any control points. This logically led to the deformation of the models (Figures 24–26).

For the Mansouria village case, the ZEB-REVO was used. Two measurements of the roofed village were performed. It was a large residential complex, documented inside and out. We also encountered deformation of the models during data processing.

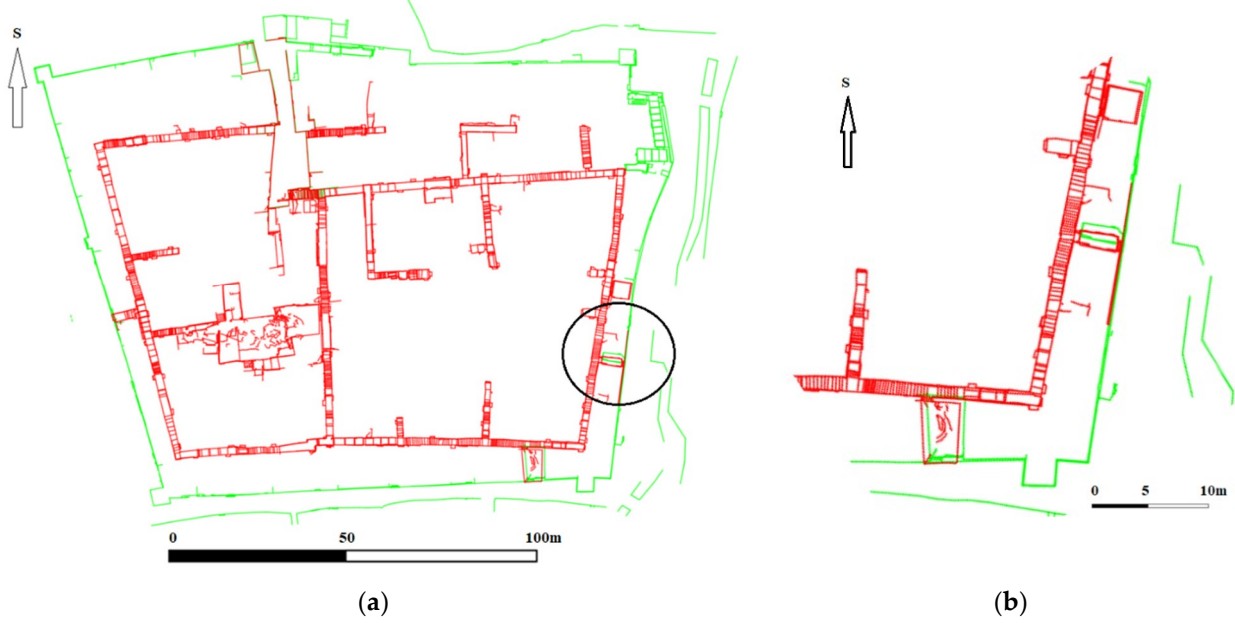

(**a**) (**b**)

**Figure 24.** After joining both models from ZEB-REVO (**a**), discrepancies of about 1.2 m occurred in the lower right part of the model (**b**).

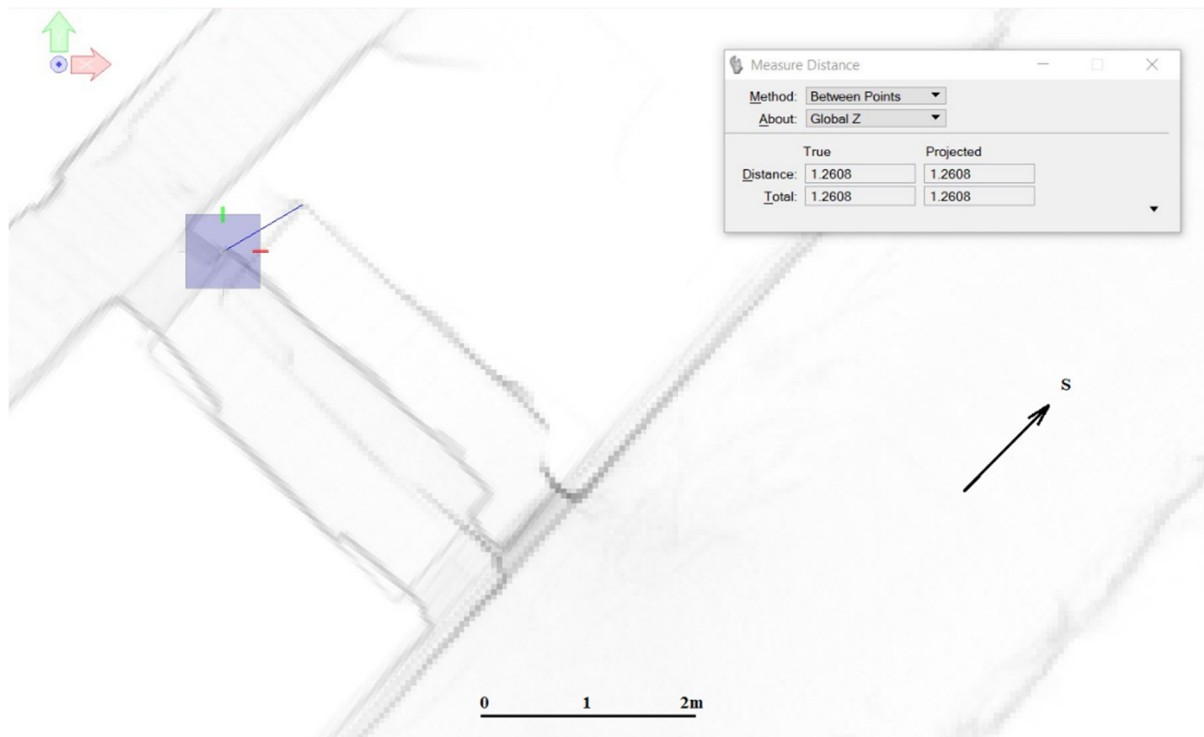

**Figure 25.** Deformation of both models.

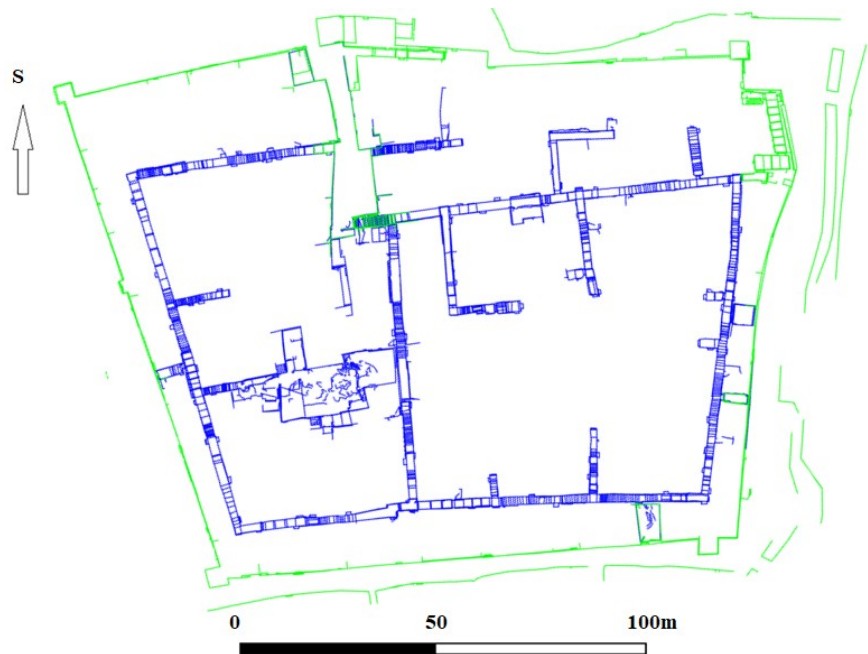

**Figure 26.** After transformation of both models from vectorized scans, an appropriate ground plan was made.

Both point clouds were then merged into one, but there was a significant discrepancy of up to 1 m. It was due to the accumulation of errors in IMU. To improve the result, both models had to be connected by a transformation based on tie points. The result is enough for this project; it is a settlement of unfired clay blocks.

In ArcGIS, the models were therefore transformed by affine transformation to 30 tie points with a resulting error of up to 15 cm, which is already acceptable. The deformation of both models caused the non-unification of data in identical areas. They were

intentionally measured during the scanning, as an unknown deformation was expected. Data merging must be done using identical scanned parts in both models (Figure 26).

The ZEB-REVO is sufficiently accurate for common documentation, especially of historical, irregular, and difficult-to-define objects (such as unfired brick objects). However, in terms of comparing accuracy and processing, it is significantly more economical and faster [40].

### 7.2. Close-Range Photogrammetry

Some objects are documented by photogrammetrical technology (IBMR). In this case, there are no suitable objects for typical and traditional photogrammetrical 3D documentation using stereophotogrammetry. IBMR can be a solution, which can replace traditional photogrammetry. As this technology produces a point cloud like laser scanners, it can replace scanners in some cases, or better, complement and merge data with data captured from scanners. By documenting artifacts (construction artifacts, archaeological funds, etc.), the IBMR nowadays is very popular and simple to use. From about 0.5–2 m and with using of a modern SLR camera, the IBMR has significantly better resolution, which reaches sub-millimeter resolution and accuracy and allows for better documenting of hidden areas (Figure 27).

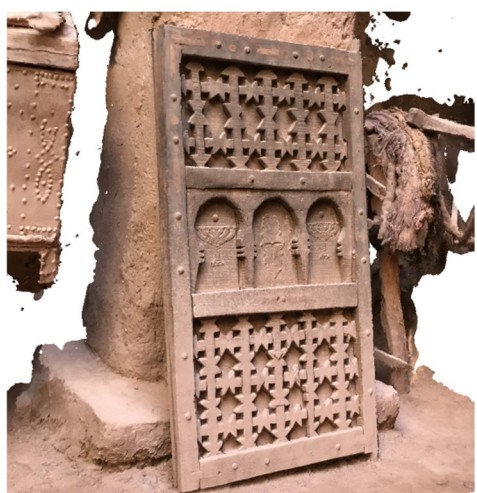

**Figure 27.** 3D model of a wooden carved artifact (Kasbah des Caids, Tamnougalt), photographed by iPhone 7 (5 min, 36 images from a distance 1–1.5 m). In this case, the created model using close-range photogrammetry was very good.

During our expedition, many artifacts were documented with this method. Important artifacts, such as archaeological finds or non-transportable objects, can then be printed as a facsimile on a 3D printer. It is also perfect that photogrammetric data can already be routinely combined with laser scanning using suitable software (e.g., RealityCapture). Thus, the advantages of both technologies can be used together, and the emerging model is suitably complemented by the second technology.

### 8. Conclusions

For typical objects made of unfired bricks and blocks, often damaged and very irregular, SLAM technology is very useful and economical. In this project, the high accuracy of object points in millimeters (like it is typical with the TLS technology) was not necessary because of shape and material used from unfired earthen blocks. Most important was the speed of the documentation of the large and complex mapped units. As was written, the accuracy of the ZEB-REVO PLS was in cm and the RMS reached was typically from 2–3 cm based on the type of measured objects. The advantages are as follows: Simple and inexpensive transport, easy operation, and automatic processing into

basic outputs. The existing software mainly creates a floor plan of a point cloud, which can be semi-automatically vectorized and saved in CAD formats (dwg, dxf). It is also possible to join individual models in one. Semi-automatic or fully automatic vectorization for irregular objects is complicated; there is a necessary, often essential, contribution from the operator. The simplest type of ZEB-REVO does not have a camera nor a display that would show what is being scanned—higher versions have it.

The BLK360 scanner is perfect for the detailed and textured scanning of objects, especially interior spaces, or small alleys, of which there are many in the historical centers of settlement. It is incredibly easy to operate with only one button. Stop-and-go scanning is relatively fast; the problem is that you usually do not see immediately how the result looks. Control via tablet and direct automatic data processing is only possible for small and simple objects; in practice, it does not work very well. Batch processing requires relatively slow (only wireless) data transfer. The automatic connection of scans is based on correlation (it does not normally need targets), and it is functional only for some objects and requires a workstation. Nevertheless, the BLK360 scanner is an amazing advancement compared to the operation and data processing of other scanners. A novelty in 2020 is the modernization of the BLK360 into a mobile scanner (e.g., ZEB-REVO). The resulting accuracy reaches 3–4 mm on short distances up to 10 m, which is precise enough for most applications of historical constructions documentation. The automatic point cloud joining runs very well for simple objects and the joining accuracy reaches 3 mm for irregular earthen structures.

Digital close-range photogrammetry is low-cost and can be the most accurate, but there are stability problems in poorly lit areas and narrow corridors. In the daytime, the outer parts of buildings are variably lit by the Sun, so it is necessary to wait for better lighting without shadows. Even sparse vegetation near the object is opaque for this technology. On the other hand, it is cheap and uses a better digital camera, which is not problematic for transport abroad. Photography is usually allowed almost everywhere, but the use of laser scanners or other professional activities can be an administrative problem in many countries. In this case, photogrammetry was used only in addition due to lack of time and poor lighting conditions. The camera used and the picture taking was not of sufficient quality; for this reason, the accuracy of the photogrammetrical result in this case reached only centimeters and the data were significantly affected by the image noise. Some photographs could not be processed due to poor lighting and poor image quality. It follows that photogrammetric work requires quality preparation and suitable lighting, which can be very demanding for indoor spaces.

In general, it is possible to recommend both of the above technologies (laser scanning generally and close-range photogrammetry), and it must be said that they significantly increase labor economics and bring exact results. They are independent of the person performing the documentation, especially compared to conventional documentation using measuring-tape or digital rangefinder and hand drawing. Thanks to using a combination of technologies, it was possible to gather exact data, which will be used for further research, leading to a more thorough understanding of southern Moroccan synagogue and *mellah* typology. Due to the character of the documented earthen towns (high urban density, irregular street system, high percentage of roofed street), it is insufficient to use aerial photos. However, the data produced by laser scanning made the further analysis of the *mellah* and its structure possible. Based on those measurements, it will soon be possible to examine the structure of the Jewish quarter in detail, to analyze its social and cultural functions, and to compare it to other quarters of a particular town. Further study on this topic will enable defining typical features of Jewish settlements in South Morocco and increase the attention of scholars and the public, which hopefully will lead to the preservation of this unique architectural and cultural phenomena.

**Author Contributions:** Conceptualization, K.P. and T.S.; methodology, K.P. and K.P.J.; technical advisory and supervision, K.P. and E.M.; data processing, K.P., K.P.J., and E.M.; investigation, K.P.

and T.S.; writing—original draft preparation, K.P., T.S., and E.M.; writing—review and editing, E.M.; visualization, K.P. and K.P.J.; project administration, K.P. and T.S. All authors have read and agreed to the published version of the manuscript.

**Funding:** This research was funded partially by the Grant SVV 2020—260484 realized at the Charles University, Faculty of Arts, partially by the Czech Technical University in Prague Grant number SGS20/053/OHK1/1T/11, and the APC was funded by the Czech Technical University in Prague, FCE, dept. of Geomatics".

**Institutional Review Board Statement:** Not applicable.

**Informed Consent Statement:** Not applicable.

**Acknowledgments:** The authors would like to thank the Charles University, Faculty of Arts, and the Faculty of Civil Engineering, CTU in Prague for support.

**Conflicts of Interest:** The authors declare no conflict of interest. The funders had no role in the design of the study; in the collection, analyses, or interpretation of data; in the writing of the manuscript, or in the decision to publish the results.

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
