# Peer review of "Earthen Jewish Architecture of Southern Morocco: Documentation of Unfired Brick Synagogues and Mellahs in the Drâa-Tafilalet Region"

_applsci, doi:10.3390/app11041712_

Round 1
Reviewer 1 Report
The manuscript presents a holistic approach towards the 3D documentation of two Jewish Cultural Heritage sites of Morocco: the Unfired Brock Synagogues and Jewish quarters (Mellahs) in Draa-Tafilalet region. The work, part of an academic ongoing project oriented to the residential areas and historic buildings of Moroccan Jews, is conducted by BLK360 and ZEB-REVO mobile laser scanners as well as the camera of an iPhone.
The content is interesting and understandable by non-specialists. While the approach is rather technical, the main part of the manuscript (introduction) elaborates on the historical context, the folklore traditions, the relations between the ethnic groups and the touristic impact. In this term, the academic contribution seems weak, and several aspects of the paper need to be improved before consider it for publication. A major revision in this sense is suggested. Here is a list of points that should be reconsidered:
General Comments
- The literature review concerning relative 3D documentation endeavors on CH historic buildings and sites, is rather poor/inadequate. Consider to include a short section or extend the Introduction. Examples of up-to date research on the topic are:
Alshawabkeh, Y., El-Khalili, M., Almasri, E., Bala’awi, F., & Al-Massarweh, A. (2020). Heritage documentation using laser scanner and photogrammetry. The case study of Qasr Al-Abidit, Jordan. Digital Applications in Archaeology and Cultural Heritage, 16, e00133. https://doi.org/10.1016/j.daach.2019.e00133
Hassan, A. T., & Fritsch, D. (2019). Integration of Laser Scanning and Photogrammetry in 3D/4D Cultural Heritage Preservation – A Review. International Journal of Applied Science and Technology, 9(4), 16. https://doi.org/ 10.30845/ijastv9n4p9
Murtiyoso, A., Grussenmeyer, P., Suwardhi, D., & Awalludin, R. (2018). Multi-Scale and Multi-Sensor 3D Documentation of Heritage Complexes in Urban Areas. ISPRS International Journal of Geo-Information, 7(12), 483. https://doi.org/10.3390/ijgi7120483
Aicardi, I., Chiabrando, F., Maria Lingua, A., & Noardo, F. (2018). Recent trends in cultural heritage 3D survey: The photogrammetric computer vision approach. Journal of Cultural Heritage, 32, 257–266. https://doi.org/10.1016/j.culher.2017.11.006
Remondino, F., & Rizzi, A. (2010). Reality-based 3D documentation of natural and cultural heritage sites—Techniques, problems, and examples. Applied Geomatics, 2(3), 85–100. https://doi.org/10.1007/s12518-010-0025-x
- The technical knowledge base of the manuscript should be further strengthened in order to outline the academic contribution of the work presented. The novelty seems to rely exclusively on the fact that the selected area of interest is not yet documented and it is a site of great importance. While the introduction includes the necessary details to fully understand the case studies and their significance, it is rather dense and over-analytical. Furthermore, the long theoretical descriptions do not align to the technical scope of the journal.
- The goal of the paper is outlined in the abstract but not in the body text. Usually, the aim of the study should appear at the end of the Introduction section.
- The authors should consider illustrating their workflow and/or the derived results with a figure or flow-chart.
- The methodology is unclear since the relative section is missing. Since the workflow includes automatic and semi-automatic techniques with no spatial reference, the proposed approach seems inappropriate to accomplish the goal of the study. This observation deserves further discussion since the readers may draw the same conclusion.
Introduction
Lines 90 – 135: Consider to remove some of the information provided concerning the cultural value of the Jewish historic buildings and presenting more concisely the footprint of the tourism.
A reference to the argument, the aim and the contributions of the work presented in this section would be strongly suggested.
Section 2
I think that the subject matter described in the lines 183 – 196 is not semantically relevant to the scope and the brief description of the project. Consider to integrate it to the next section in order to justify the choice of the used equipment and methodology for data collection. Moreover, Section 2 could be part of the Introduction.
Section 4
Line 238: I suggest to omit the sentences referring to data transferring as this procedure cannot be considered as a part of the workflow.
Line 257: Please clarify the procedures the editing includes.
Line 260: The photogrammetric software packages mentioned as well as the free (open-source) ones are considered as a necessary adjunct to the literature review of the manuscript. Reference papers should be added to supplement the literature review, such as:
Jaafar, H. A., Meng, X., Sowter, A., & Bryan, P. (2017). New approach for monitoring historic and heritage buildings: Using terrestrial laser scanning and generalised Procrustes analysis. Structural Control and Health Monitoring, 24(11), e1987. https://doi.org/10.1002/stc.1987
Pritchard, D., Sperner, J., Hoepner, S., & Tenschert, R. (2017). Terrestrial laser scanning for heritage conservation:the Cologne Cathedral documentation project. ISPRS Annals of Photogrammetry, Remote Sensing and Spatial Information Sciences, IV-2/W2, 213–220. https://doi.org/10.5194/isprs-annals-IV-2-W2-213-2017
Bartoš, K., Pukanská, K., & Sabová, J. (2014). Overview of Available Open-Source Photogrammetric Software, its Use and Analysis. International Journal for Innovation Education and Research, Vol. 2(10). https://doi.org/10.31686/ijier.vol2.iss4.170
Grussenmeyer, P., & Khalil, O. A. (2008). A comparison of photogrammetry software packages for the documentation of buildings. Halshs, 9.
Line 263: Consider to omit or rephrase the sentence: With a 60% - 80% overlap between the images and ensuring that all important areas of the object are visible in at least three images, there is “hope” that a dense point cloud with the useful 3D information will be produced. A proper strategy during capturing the image data will guarantee a geometrically accurate result without missing spots. IBMR often outperforms laser scanning (as you mention in line 398), which apparently is subject to its own restrictions and difficulties.
Section 5:
Line 280: Please mention the factors that determine the quality of the produced 3D model as they are described in the caption of Figure 10. (sparse point cloud leading to holes – information missing, problems with texturing, low-resolution in texture mapping, metric errors, distortion etc.)
Line 281: Consider rephrasing cause it is misleading for non-experts. I do not think that this statement is an inference for the efficacy of close-range photogrammetry; it was expected considering the use of a smartphone and the conditions and it does not prove anything. Moreover, the Section 6.2 opposes this argument.
Line 283: ‘fatal consequences’ can be replaced by ‘inadequate/ insufficient results’
Section 6:
Line 387: Please justify why the result is very good. A figure from CloudCompare with the deviation comparison/ distance mapping could enhance this point.
Conclusion:
Line 412: Consider to rephrase the sentence.
Line 414: Verb is missing.
Author Response
Please, find the attachment. Many thanks for your comments.

Reviewer 2 Report
In the Introduction the analysis of state-of-the-art literature in this field needs to be presented. Please analyze the devices used in this field, their advantages and disadvantages, reported accuracy for similar studies, etc. The aim of the study needs to be presented. Please provide the accuracy requirements.
Please provide the dimension of the synagog and the plan of measurement i.e. scanner positions. How did you match BLC and ZEB-REVO point clouds?
Would be more appropriate to use sphere targets for joining models in dark spaces? Why didn't you use them?
Please compare your results with similar studies.
Author Response

(The authors gave the same response as above.)

Reviewer 3 Report
General considerations
The paper describes a survey conducted in Morocco for documenting a Jewish Architecture. The proposed methodology employs several approaches like a miniatured TLS (Terrestrial Laser Scanning), a handheld scanner based on SLAM approach and the close photogrammetry as well. This work is part of a project of Czech Technical University and Charles University in Prague and the aim is a survey of the Jewish architecture of Southern Morocco.
Of course, the topic is suitable for this journal, but the paper describes the historical part in deep, letting a weak description of the survey and the obtained results. Although the portion of the paper dedicated to the survey methodology is too short, the paper is well organized in six sections. The length of some sections is not reasonable, the introduction is too long, while the sections 3 and 4 can be enlarged. The title is misled; indeed, the word “location” creates an expectation for the reader about something like a georeferencing, that is not present in the paper.
The abstract is readable and clear when the authors write “analyses of accuracy”, personally, I was looking forward to seeing somethings very specific about this issue.
The introduction deals with historical and anthropological background, for a paper presented in “Applied Science” such a section is too long. Moreover, the bibliography is very rich and takes more than the 60% (19/31) of the entire paper. I prefer that for a paper in “Applied Science” the scientific reference must cover the 50% at least. Finally, the figure 1-2-3 are very evocative but there are not reference in the text.
The section 2 provides a sufficient description of the project and the length is reasonable but I have some doubts about the reference [26]. The section 3 is sufficient clear, I suggest to cite the original paper of SLAM. The section 4 should report the post-processing in deep and the method used for estimate the accuracy and precision of the surveys. The section 5 provides many images but is scientifically lack, you cannot use a TS (Total Station) but you could use targets (volumetrics and planar) to join the surveys as well as to limit the deformation. Simple scale-bar also could mitigate the deformation of the model especially for the photogrammetric acquisition.
The conclusions are appropriated and coherent with the paper but can be extended with further considerations.
The number of references is not sufficient.

Author Response

(The authors gave the same response as above.)

Round 2
Reviewer 1 Report
Thank you for addressing my comments by (i) revising the structure of the manuscript, (ii) adjusting the contents of Introduction and Implementation sections, (iii) including the proposed references and finally, (iv) illustrating the presented work in a Figure. The work is presented with a good clarity of style and no further revisions are needed.
Author Response
Dear reviewer, ok, thank you very much. After regests of next reviewers the article was changed with added accuracy statistics in information.
Reviewer 2 Report
The authors provided an improved version compared to the first round. Therefore, in my opinion, this paper has great potential for publication in this journal. However, in my opinion, several improvements need to be made:
-Although this research has potential, its current presentation is a more technical report than a scientific paper. The scientific soundness needs to be improved.
-The parts that contain information about projects and historical information need to be reduced.
-Although the authors added the review of used technology in this field, they didn't provide a deeper analysis of suggested methodologies, and they shourtcoimngs therefore it is difficult to understand the main aim of this studies. Several papers comparing different geospatial technologies for documentation of cultural heritage have been published. How your research is different?
- The accuracy assessment and validation of the presented approach and application of technology are missing. It is necessary to provide a comparative analysis of presented technologies and provide an accurate assessment. Please provide a deep discussion of cloud compare results. Also, you can extract cross-sections on specific locations and comment on differences.
-The paper needs to be restructured especially the abstract and result and discussion section. The results need to be commented by using facts, not descriptions, - the model used for accuracy assessment (relative between used technologies or absolute from the point of view of the results) should be presented -Table 1 and Table 2 aren't based on results. They contain the characteristics of used technologies. Therefore they don't belong to the result and discussion section.Author Response
Dear reviewer, thank you very much. Please find the changes in attached pdf in red colour.
To your remarks / recommendation:
The authors provided an improved version compared to the first round. Therefore, in my opinion, this paper has great potential for publication in this journal. However, in my opinion, several improvements need to be made:
-Although this research has potential, its current presentation is a more technical report than a scientific paper. The scientific soundness needs to be improved.
Ok, we add a testing of new used instruments which was made directly before the expedition. Both devices are purchased in late 2019 in firstly used in field measurement in Morocco in February 2020. Next, in the Case project paragraph other information regarding accurycy were added.
-The parts that contain information about projects and historical information need to be reduced.
It is difficult to reduce the historian information, it was reduced after first review round from 3 pages to one page. This article (and research) was based on collaboration with Faculty of Philosophy and History of the Charles University; the researchers and students from this faculty prepared the expedition. It is necessary for their activity to write some sentences about the project financed from university student grant.
-Although the authors added the review of used technology in this field, they didn't provide a deeper analysis of suggested methodologies, and they shourtcoimngs therefore it is difficult to understand the main aim of this studies.
The main aim of this project was to test quick and easy to transport modern laser scanning technology in the Morocco for little known historical monuments threatened by imminent destruction. Both instruments are relatively new and for researchers in culture, history and monuments care are unknown. Personal mobile laser scanners (PLS) are still under development and are not a common devices. Therefore, I think, their use and testing is interesting for many experts; it was focused not only for technicians, but for resercbhers and workers in monuments care.
Several papers comparing different geospatial technologies for documentation of cultural heritage have been published. How your research is different?
Yes, several were published, but the technology of PLS is releticelly new and under developing, such as miniaturising of measuring instruments. Testing of their accuracy and usability in cultural heritage documentation in real practice is the benefit of our research (testing - accuracy, price, transportation, time demend, easy to use, data volume etc.). A second aim is an information about abandoned cultural places which are on the edge of interest and in danger of their destruction, which corresponds with focusing of the grant financed from Charles university, Faculty of history and art.
- The accuracy assessment and validation of the presented approach and application of technology are missing. It is necessary to provide a comparative analysis of presented technologies and provide an accurate assessment. Please provide a deep discussion of cloud compare results. Also, you can extract cross-sections on specific locations and comment on differences.
OK, we add a part, which describes the laboratory (at the faculty) testing. Cross sections were added to the case project „synagogue „which show the quality of captured point clouds. There is a possibility to confront the data-noise from different technologies, which affects results quality.
-The paper needs to be restructured especially the abstract and result and discussion section. The results need to be commented by using facts, not descriptions, - the model used for accuracy assessment (relative between used technologies or absolute from the point of view of the results) should be presented -Table 1 and Table 2 aren't based on results. They contain the characteristics of used technologies. Therefore they don't belong to the result and discussion section.
OK, you have right with tables; restructured, tables were located in other part of the text, next tables with accuracy result were added. Abstract, conclusion and discussion were partial changed.

Reviewer 3 Report
2nd round
The paper describes a survey conducted in Morocco for documenting a Jewish Architecture. The proposed methodology employs several approaches like a miniatured TLS (Terrestrial Laser Scanning), a handheld scanner based on SLAM approach, and close photogrammetry as well. This work is part of a project of Czech Technical University and Charles University in Prague and the aim is a survey of a Jewish architecture of Southern Morocco.
Thank you to the authors for all edits performed. The new organization of the paper sounds good. Now the technical and scientific soundness is improved, even if could be improved further reporting the RMSE local and global of the cloud point alignment.
Just two further issue to solve:
- Please review the workflow: in the first workflow, the word Hub is partially hidden. In the second one, you can split the data processing highlighting the alignment procedure using ICP (essential for this application), for this reason, if you need to collect space you can reduce the “Data Export via…” in Data Export simply.
- The new sentence 856-857 “point clouds differ only in parts that are not common” sounds bizarre, if the part is not in common you cannot compare them. I suggest to review this.
Finally, the geomatic part is still weak, no statistical parameters are reported. I agree with the authors that this article should be for applied sciences (cit. Cover Lecter), but it is not a simple workshop. Of course, if you did not use targets to evaluate the deformations or other innovative methodologies, now you cannot come back to perform it.
Author Response
Dear reviewer, thank you very much. Please find the changes in attached pdf in red colour.
To your remarks / recommendation:
The paper describes a survey conducted in Morocco for documenting a Jewish Architecture. The proposed methodology employs several approaches like a miniatured TLS (Terrestrial Laser Scanning), a handheld scanner based on SLAM approach, and close photogrammetry as well. This work is part of a project of Czech Technical University and Charles University in Prague and the aim is a survey of a Jewish architecture of Southern Morocco.
Thank you to the authors for all edits performed. The new organization of the paper sounds good. Now the technical and scientific soundness is improved, even if could be improved further reporting the RMSE local and global of the cloud point alignment.
OK, thank you, some new information about precision were added after your and next reviewer request
Just two further issue to solve:
- Please review the workflow: in the first workflow, the word Hub is partially hidden. In the second one, you can split the data processing highlighting the alignment procedure using ICP (essential for this application), for this reason, if you need to collect space you can reduce the “Data Export via…” in Data Export simply.
OK, changed
|
- The new sentence 856-857 “point clouds differ only in parts that are not common” sounds bizarre, if the part is not in common you cannot compare them. I suggest to review this.
OK, changed, you have right.
This was meant so, that some parts of the point cloud from the BLK or from photogrammetry were missing. It was mainly a cult place behind the wall for storing books. The hand-held scanner could be inserted there, other technologies did not allow it.
Finally, the geomatic part is still weak, no statistical parameters are reported. I agree with the authors that this article should be for applied sciences (cit. Cover Lecter), but it is not a simple workshop.
OK, we added next statistical parameters.
Of course, if you did not use targets to evaluate the deformations or other innovative methodologies, now you cannot come back to perform it.
Yes, we didnt’t use special targets and geodetical measurement using total station. We assume, that the precision of the BLK360 scanner is much better than ZEB-REVO, which has precision 1-3cm depend on material and scanning process. The BLK360 has a precision 4mm on 10m. All distances were under 8m. It means, we used the point cloud from the BLK360 as reference one.
In the text we add an information about the data noise, which is by the ZEB-REVO large.

Round 3
Reviewer 2 Report
Thank you for your patience and hard work.